# Structure-energy-based predictions and network modelling of RASopathy and cancer missense mutations

Christina Kiel[1,2,*] & Luis Serrano[1,2,3]

## Abstract

The Ras/MAPK syndromes ('RASopathies') are a class of developmental disorders caused by germline mutations in 15 genes encoding proteins of the Ras/mitogen-activated protein kinase (MAPK) pathway frequently involved in cancer. Little is known about the molecular mechanisms underlying the differences in mutations of the same protein causing either cancer or RASopathies. Here, we shed light on 956 RASopathy and cancer missense mutations by combining protein network data with mutational analyses based on 3D structures. Using the protein design algorithm FoldX, we predict that most of the missense mutations with destabilising energies are in structural regions that control the activation of proteins, and only a few are predicted to compromise protein folding. We find a trend that energy changes are higher for cancer compared to RASopathy mutations. Through network modelling, we show that partly compensatory mutations in RASopathies result in only minor downstream pathway deregulation. In summary, we suggest that quantitative rather than qualitative network differences determine the phenotypic outcome of RASopathy compared to cancer mutations.

**Keywords** RASopathy; FoldX; MAPK pathway; missense mutations; enedgetics
**Subject Categories** Network Biology; Molecular Biology of Disease; Signal Transduction
**Mol Syst Biol.** (2014) 10: 727

## Introduction

RASopathies are a group of germline developmental disorders of the Ras-MAPK pathway, such as Noonan, cardio-facio-cutaneous (CFC), Costello and LEOPARD syndromes (Tartaglia & Gelb, 2005; Bentires-Alj *et al*, 2006; Schubert *et al*, 2007b; Aoki *et al*, 2008; Tartaglia *et al*, 2010). These rare diseases – with one per 500–2,500 individuals annually affected – share phenotypic features that include postnatal reduced growth, facial dysmorphism, cardiac defects, mental retardation, skin defects, musculo-skeletal defects, short stature and cryptorchidism (Supplementary Fig S1). Most mutants described result in up-regulating the RAS-RAF-ERK-MAPK-kinase cascade (Tartaglia & Gelb, 2005; Rodriguez-Viciana *et al*, 2006; Roberts *et al*, 2007; Tartaglia *et al*, 2007; Pandit *et al*, 2007; Sarkozy *et al*, 2009; Lepri *et al*, 2011; Andreadi *et al*, 2012) and are found in 15 genes: PTPN11, SOS1, RASA1, NF1, KRAS, HRAS, NRAS, BRAF, RAF1, MAP2K1, MAP2K2, SPRED1, RIT1, SHOC2 and CBL (Aoki *et al*, 2013; Rauen, 2013). These 15 genes form a connected network with no isolated members (Fig 1A): CBL proteins have protein tyrosine kinase (e.g. EGFR)-directed E3 ubiquitin ligase functions, which then promote the degradation of substrates by the proteasome (Nadeau *et al*, 2012). PTPN11 (gene product SHP2) is a phosphatase that is recruited through an SH2 domain binding to receptor tyrosine kinases (RTKs). Son of sevenless 1 (SOS1) is a nucleotide exchange factor (GEF) that gets recruited to the membrane in response to growth factor receptor activation, and catalyses the activation of Ras proteins (Vetter & Wittinghofer, 2001). The Ras protein family with the isoforms H-, K- and N-Ras (highly similar in sequence) plays a central role in the Ras-MAPK signalling cascade. Rit1 is a Ras family member that activates BRAF/MAPK and p38 kinase signalling (Shi & Andres, 2005; Aoki *et al*, 2013; Berger *et al*, 2014). Ras proteins bind the guanine nucleotides GDP and GTP tightly and act as molecular switches through cycling between an inactive (GDP-bound) and an active (GTP-bound) state (Vetter & Wittinghofer, 2001). In the active form, Ras interacts with effectors, such as the serine-threonine kinases RAF1 and BRAF, and thereby stimulates downstream activation of the MEK (MAP2K1/MAP2K2; MEK1/2)-ERK pathway. The GAP (GTPase activating) proteins include RASA1 (p120 RasGAP) and NF1 (protein neurofibromin-1), which are crucial for down-regulating Ras activation by catalysing the slow intrinsic GTP hydrolysis of Ras. SPRED1 prevents RAF1 activation by Ras (Brems *et al*, 2007). SHOC2 is a scaffold that positively regulates Ras-effector signalling (Cordeddu *et al*, 2009; Kaplan *et al*, 2012; Young *et al*, 2013).

It is intriguing that mutations in the same 15 genes are also frequently identified in different types of human cancers (Fig 1B). In

1   EMBL/CRG Systems Biology Research Unit, Centre for Genomic Regulation (CRG), Barcelona, Spain
2   Universitat Pompeu Fabra (UPF), Barcelona, Spain
3   Institució Catalana de Recerca i Estudis Avançats (ICREA), Barcelona, Spain
    *Corresponding author. Tel: +34 93 316 02 59; Fax: +34 93 316 00 99; christina.kiel@crg.eu

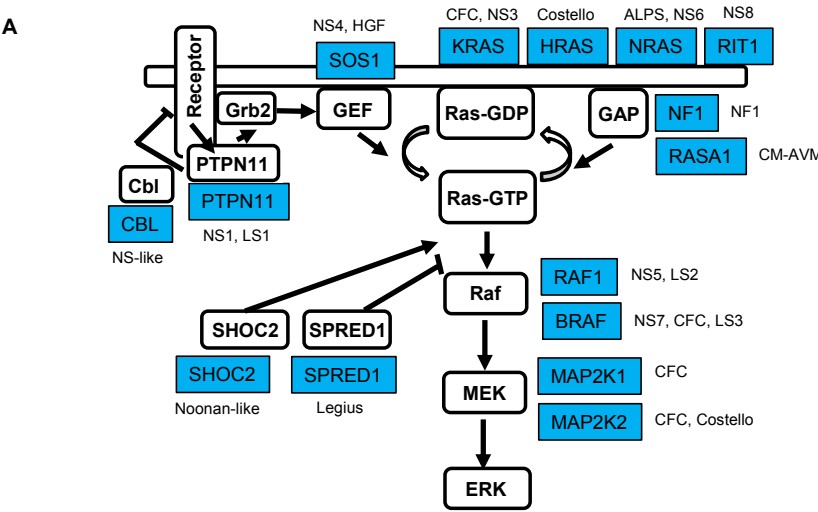

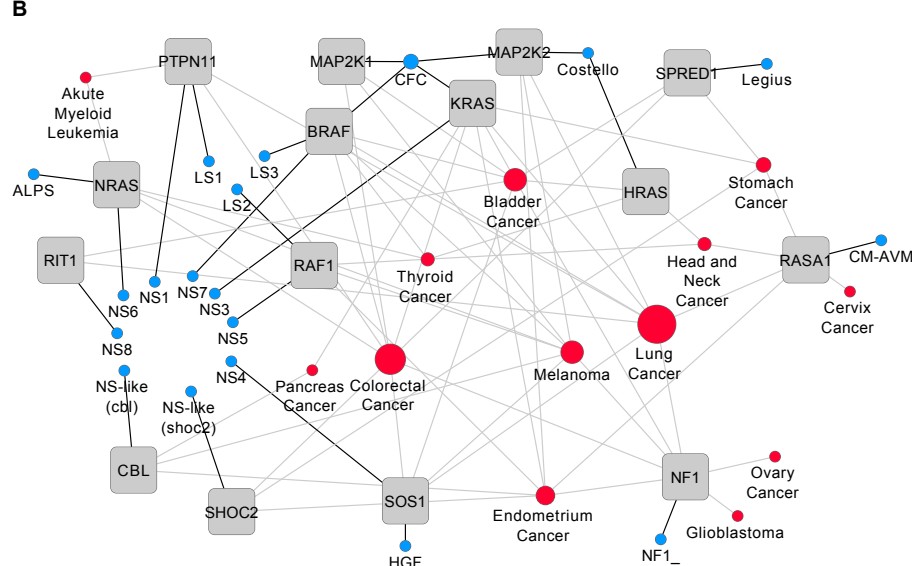

**Figure 1. Genes affected in RASopathies.**

A   Network diagram showing affected genes in RASopathies. Proteins (PTPN11, SOS1 and SPRED) and protein groups (Ras, including NRAS, HRAS, KRAS and RIT1; GAP, including NF1 and RASA1; Raf, including RAF1 and BRAF; MEK, including MAP2K1 and MAP2K2) are displayed in white boxes and arranged in a network with their respective genes in grey. RASopathy diseases are indicated in blue.

B   Diseasome of RASopathies and cancer. Each node corresponds to a distinct disorder or cancer type. The size of the node corresponds to the total number of genes (among the 15 genes) that are involved in a particular disease. Abbreviations: NS, Noonan syndrome; NF1, neurofibromatosis type 1; CFC, cardiofaciocutaneous; LS, LEOPARD syndrome; HGF, hereditary gingival fibromatosis; CM-AVM, capillary malfunction-arteriovenous malfunction; ALPS, autoimmune lymphoproliferative syndrome. Suffixes in NS (NS1, NS3, NS4, NS5, NS6, NS7, NS8 and NS-like) and LS (LS1 to LS3) are different forms of the respective disease according to the classification in the OMIM database.

some cases, cancer-associated somatic mutations result in much stronger increased signalling along the Ras-MAPK pathway measured on ERK (Tartaglia *et al*, 2003; Keilhack *et al*, 2005; Gremer *et al*, 2011). For example, Tartaglia and colleagues assessed PTPN11 (SHP2) mutations by comparing somatic mutations in juvenile myelomonocytic leukaemia (JMML) with Noonan syndrome germline mutations, and they identified a greater activation of the RAS/MAPK activation for the JMML-associated mutations (Tartaglia *et al*, 2003). Whether this trend holds for all RASopathies has not been systematically demonstrated yet.

Mapping of disease-related missense mutation onto different areas of protein structures has led to valuable insights into the molecular mechanism underlying a respective disorder. Examples are the proposed release of autoinhibition in PTPN11 and SOS1, which render the proteins constitutive active (Gureasko *et al*, 2008; Araki *et al*, 2009; Tartaglia *et al*, 2010). However, only little insights into the quantitative relation between the impact of a mutation and its structural localisation and disease severity have been achieved for proteins involved in RASopathies and/or cancer (Keilhack *et al*, 2005; Tartaglia *et al*, 2006; Cirstea *et al*, 2010; Molzan *et al*, 2010).

Structure-based protein design algorithms, such as FoldX (Schymko-witz et al, 2005; Van Durme et al, 2011), can be used to predict the energetic impact of a mutation on protein stability or complex stability (Alibes et al, 2010; Simoes-Correia et al, 2012). For genetic disorders that are mainly caused by a decreased protein stability, such as phenylketonuria or retinitis pigmentosa, high correlations between the onset of the disease and the destabilisation (unfolding) of a mutation based on FoldX energies have been demonstrated (Pey et al, 2007; Rakoczy et al, 2011).

In this study, we systematically analysed the mutations occurring in the same proteins that result in RASopathies or cancer, to find whether there are common trends (for some proteins or structural regions) that could reinforce the hypothesis of weaker deregulation of the RAS/MAPK pathway in RASopathies compared to cancer. We analysed 956 different missense mutations for the 15 proteins involved in RASopathies and/or cancers using available 3D structures, energy calculations, sequence-based tools and known experimental information. We find that even for the same gene, depending on the type of the mutation, different disease-causing mechanisms exist. Through our analysis, we observed the trend that energy changes on average are higher for cancer compared to RASopathy mutations. Finally, RASopathy mutations show in some cases compensatory changes that by network modelling are predicted to result in a smaller pathway deregulation. Altogether, our study features the relevance of including quantitative edge effects (affinities and kinetic constants) in systems approaches that integrate tissue or patient-specific protein abundances with disease networks.

## Results

### Participation in 33 signalling pathways

We first investigated in which signalling pathways the 15 RASopathy proteins participate by using the NetPath/NetSlim database (a manually curated resource that lists 33 signalling pathways; Kandasamy et al, 2010). As RASopathy genes are involved in multiple pheno-typical disorders with many diverse and overlapping and clinical symptoms (Supplementary Fig S1), we expected that the 15 proteins are participating in most if not all signalling pathways. We first analysed for all 1,816 signalling proteins in NetPath how many pathways each protein participates in (Supplementary Fig S2A). While this network for all 1,816 proteins is 'scale-free' (following a power law), with only few proteins participating in most of the pathways (such as ERK1, ERK2, AKT1 and PI3K) and many proteins only in one pathway (Supplementary Fig S2B), the opposite trend is found for the 15 RASopathy proteins (Supplementary Fig S2C). In accordance with our expectation, most of the 15 proteins are participating in a high number of signalling pathways (on average in 10 pathways). This number is likely to be higher as in NetPath not all isoforms of RAF1 and BRAF as well as of K, N and H-RAS are always listed. In fact, many of the RASopathy genes are embryonic lethal when knocked out, and thus are essential disease genes (Dickerson et al, 2011). Exceptions are HRAS, NRAS, MAP2K2; KRas4 mice are viable and only KRas4B is embryonic lethal (Umanoff et al, 1995; Plowman et al, 2003). In case of CBL, only deleting both CBL and CBLB results in embryonic lethality (Naramura et al, 2002). No

information on embryonic lethality was found for Rit1. Interestingly, other Ras-MAPK proteins participating in a very large number of pathways, such as ERK (in 29 pathways), are excluded from the list of RASopathy-related genes.

Next we analysed in more detail the function of the 33 signalling pathways. The pathways which have the most RASopathy proteins participating are downstream of the B- and T-cell receptors, the epidermal growth factor receptor (EGFR), Kit receptor, oncostatin (OSM) receptor, prolactin, fibroblast growth factor receptor (FGF-1), brain-derived growth factor receptor (BDNFR), different interleukin receptor family members, tumour necrosis factor alpha (TNF-α) receptor and transforming growth factor beta (TGF-β) (Supplementary Fig S3). All these pathways are known to mediate diverse biological functions, such as cell proliferation, survival, differentiation, and are activated in response to cytokine, hormone and growth factor stimulation. In contrast, the core embryonic developmental pathways, Hedgehog, NOTCH and Wnt, do not involve RASopathy-related proteins. Likewise, the 15 proteins are not found in the TNF-related weak inducer of apoptosis (TWEAK) pathway. Consistent with the participation in many signalling pathways, the 15 RASopathy proteins are expressed in most of the tissues (Supplementary Figs S4 and S5).

In conclusion, ubiquitously RASopathy proteins are involved in mediating the early to late developmental processes, including morphology determination, organogenesis, synaptic plasticity processes and growth, but they are excluded from very early embryonic developmental pathways. This could explain in part the overlapping RASopathy disease symptoms, as well as the morphological defects.

### Domain analysis of missense mutations in RASopathies and cancer

We obtained the list of RASopathy disorders and disease genes from the Online Mendelian Inheritance in Man (OMIM; http://www.omim.org/) database. We next compiled a list of missense mutations for RASopathies and cancer from OMIM, the Human Gene Mutation Database (HGMD), UniProt and COSMIC (see Supplementary Table S1). In total, we collected 295 Mendelian (germline) and 603 somatic mutations. In general, all RASopathie mutations are germline, while 90% of cancer mutations are somatic (Futreal et al, 2004). Only 58 mutations (6%) have been associated with both germline and cancer mutations. Thus, in the following text, when we refer to germline mutations we refer to RASopathies and somatic mutations refer to cancer. Mutations of MA2K1, MAP2K2, HRAS, NF1, SOS1, PTPN11 and RAF1 have higher fractions of germline mutations, while KRAS, CBL, NRAS, SHOC2 and BRAF have similar proportions (Supplementary Fig S6).

The enrichment of missense mutations in different Pfam domains causing different disorders has been observed earlier (Zhong et al, 2009). To see whether germline and cancer mutation fall into distinct groups with respect to enrichment in domains, we first used Pfam to predict the domains for all proteins and then mapped all missense mutations onto domains and the structural regions in-between domains (domains or inter-domain regions; Fig 2). Germline mutations do not distribute equally over the 98 possible structural regions and are absent in 57 regions. Somatic mutations are more equally distributed. In many cases (38%), the same structural

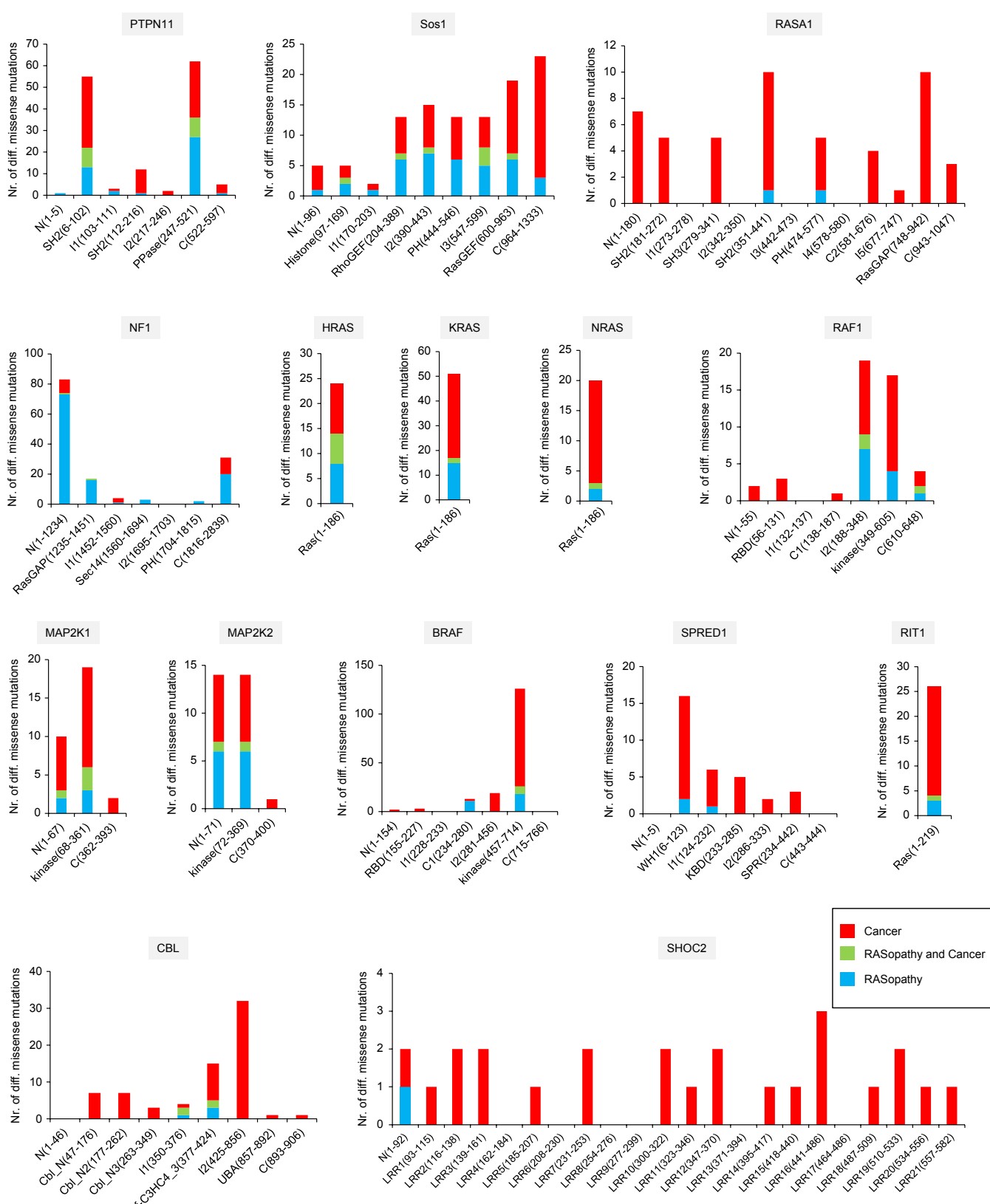

**Figure 2.  Distribution of somatic and germline mutations in 98 different structural domains and inter-structural regions.**
For each gene, the number of different missense mutations were mapped on the respective domains and inter-structural regions (called I1 to Ix, and 'N' = N-terminal region; 'C' = C -terminal region). The colour of the bar diagrams represent mutations from different classes (see legend).

region contains both cancer and RASopathy-related mutants (Supplementary Fig S7). We find enrichment in RASopathy mutations for NF1, while RASA1 and SHOC2 have mainly cancer mutations. SOS1 has both types of mutations at every structural region. In conclusion, we find that for many proteins cancer and germline mutations cannot be distinguished based on their localisation in specific protein domains.

**Classification of disease mutations and energy calculations of mutants using FoldX**

Mutations in a protein can have five possible different general effects (Fig 3A): they can alter protein activity (class 1), can affect protein-protein interactions (class 2), destabilise a globular domain preventing folding (class 3), can affect protein localisation and/or half-life (class 4) or can be neutral (class 5). Of course, it is possible to have combined effects as well. Regarding changes in protein activity, mutations can affect the active site or the ligand binding pocket of a protein thus generally impairing activity. For class 1, in the case of multi-domain proteins, or proteins with regulatory unstructured protein segments, mutations could impair domain-domain inhibitory interactions (subclass 1a), or release an inhibitory protein segment (subclass 1b), thus resulting in activation (please note that in some cases releasing domain-domain interactions could result in activation of signalling by changing protein localisation). In Fig 3B, we show relevant protein structures with the corresponding mutations for the five categories).

In order to classify all mutations according to the different classes and groups (Fig 3A), we used structural information from the protein data bank (pdb) and of complex structures (Supplementary Fig S8 and Materials and Methods), the protein design algorithm FoldX (Guerois *et al*, 2002; Schymkowitz *et al*, 2005) and experimental information from literature. In total 65% of the missense mutations (621 mutations) are covered by a 3D structure from the pdb and were analysed by FoldX (see Supplementary Table S1). We considered FoldX predicted energy changes ($\Delta\Delta G$ values) larger than 1.6 kcal/mol as highly significant (99% confidence interval), as they correspond to twice the standard deviation of the error in FoldX (Kiel *et al*, 2004; Schymkowitz *et al*, 2005; Rakoczy *et al*, 2011). Smaller energy changes of > 0.8 kcal/mol were still considered significant (one standard deviation; 95% confidence interval). Out of the 621 mutants modelled by FoldX, 427 have significant energy changes (Fig 4A and B), or they affect catalysis, or membrane localisation (17 mutants known from literature). The 427 cases include 311 highly significant mutants (> 1.6 kcal/mol; Supplementary Fig S9). Of those, 65% results in changes in inter- or intra-domain interaction energies and 35% affect domain folding energies (Fig 4B). Regarding folding mutants, 41% are in the inhibitory proteins of the network: NF1, RASA1, SPRED1 and CBL (Fig 4C). Three mutations affect localisation, based on experimental information: two directly affect membrane localisation (the histone domain in Sos1; Gureasko *et al*, 2008) and one introduces an N-myristoylation site in SHOC2 that results in aberrant targeting to the plasma membrane and results in impaired translocation to the nucleus upon growth factor stimulation (Cordeddu *et al*, 2009).

A total of 197 mutants do not have significant changes in energies within the FoldX error (< 0.8 kcal/mol), and no information about the disease-causing mechanism is known from literature.

Structural inspections revealed that 82% of the 199 mutations are located at the protein surface and may affect binding to a partner protein (Supplementary Fig S10A). For example, mutations in the SH2 domain of RASA1 may prevent binding to phosphorylated peptides (such as the one from the EGFR receptor), as seen based on homology to the SH2 domain of NCK1 in complex with a peptide (pdb entry 2CIA) (Frese *et al*, 2006). We used sequence-based methods to indicate the likelihood of being disease-causing for the 197 mutations. As a first classifier ('sequence conservation'), we analysed the evolutionary sequence conservation for amino acid positions affected by mutations, as a higher conservation is expected for disease-causing mutations ('Shannon entropy') (Strait & Dewey, 1996). A Shannon value below the mean for all disease mutation was used as the threshold for high sequence conservation (Supplementary Fig S11A). For a second classifier ('amino acid [AA] substitution diverseness'), we used BLOSUM matrix changes (Henikoff & Henikoff, 1992), with values below the mean of all disease mutants as the threshold for high AA substitution diverseness (Supplementary Fig S11B). With the combined information from BLOSUM matrixes and Shannon entropies, 20% of the non-classified mutants are likely to be disease-causing and another 40% are maybe disease-causing (Supplementary Fig S10A), while 23% are likely to be non-disease-causing based. The remaining 17% of mutations are in the hydrophobic core, of proteins, but with destabilising FoldX changes below the threshold ($\leq 0.8$ kcal/mol).

A total of 332 mutants could not be modelled on a 3D structure (Fig 4A). We performed a similar sequence-based analysis based on BLOSUM matrix changes and Shannon entropies for the 332 mutants (Supplementary Fig S10B). The analysis shows similar results as obtained for the surface mutations, with 32% of the mutants likely to be disease-causing, and another 43% that are maybe disease-causing. Twenty-five per cent are likely to be non-disease-causing based on sequence conservation.

As a summary for 14 out of the 15 proteins involved in RASopathies, groups of mutants can be assigned to different underlying disease-causing mechanisms (Fig 4C). No protein structure has been solved, and no experimental biochemical information is available for Rit1. For 12 out of 15 proteins, we find different classes of mutations in the same protein. Thus, when integrating disease networks with tissue-specific protein expression, different node and edge properties have to be considered, even for the same disease gene.

**Differences in mutation energy effects between missense mutations in RASopathies and cancer**

We next compared destabilising FoldX energies for all mutants by separately averaging RASopathy/germline and cancer/somatic mutants. The cancer mutations were extracted from the OMIM database, which lists all mutations even if not confirmed to be disease-causing [in cancer often > 200 proteins are mutated and not all are expected to be disease-causing (Vogelstein *et al*, 2013)]. To exclude possible cancer passenger mutations, we generated a 'gold set' of RASopathy and cancer mutations by including only those mutations that have been already studied experimentally, and for which there is evidence of their transforming potential, or for which sustained increased kinase activity has been demonstrated (Wan *et al*, 2004; Gremer *et al*, 2011) (Supplementary Table S2). Using this gold set, the FoldX energy values for germline and cancer mutants separate

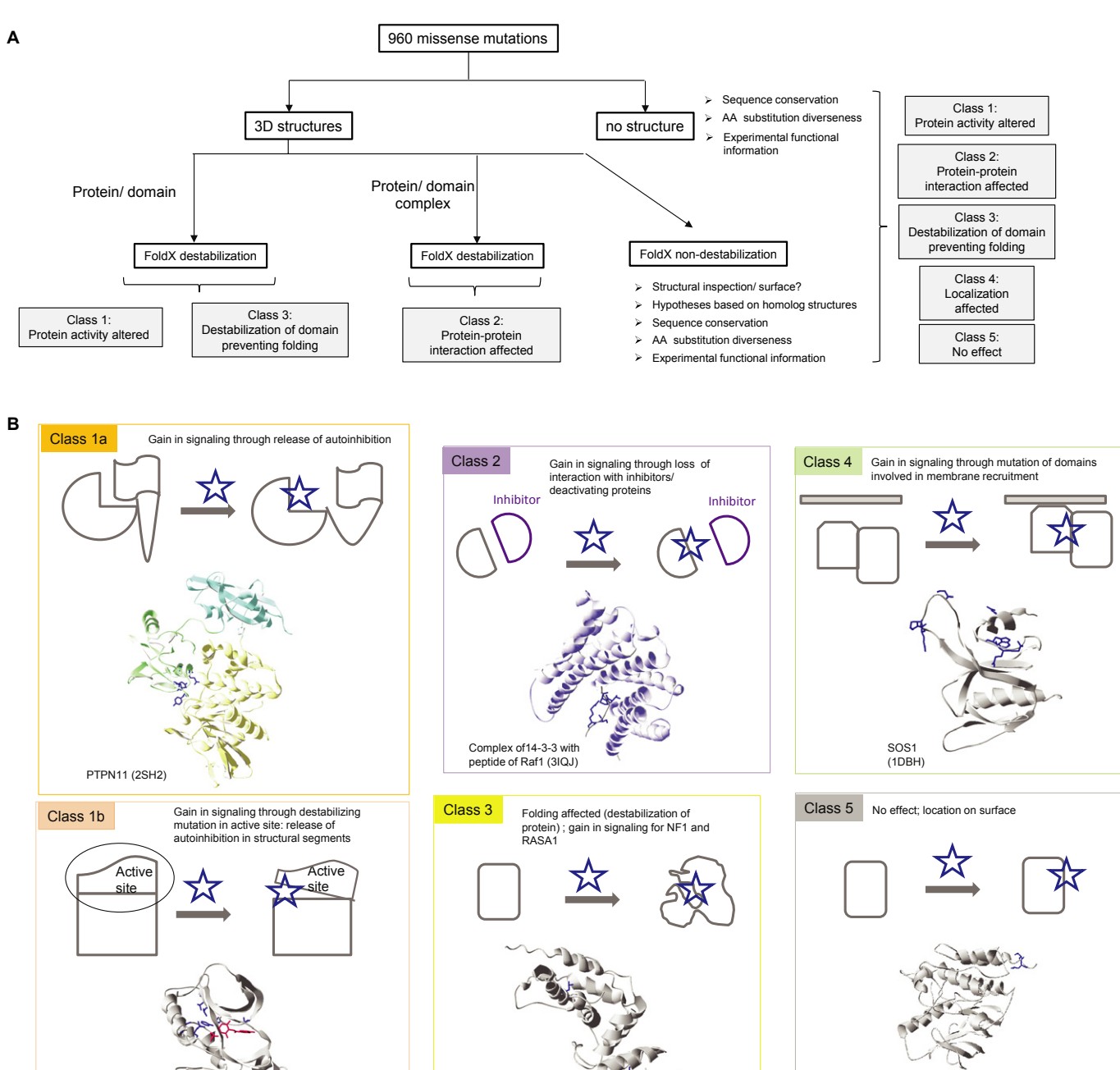

**Figure 3. Classification of missense mutations and flow chart to analyse missense mutations based on structural information and FoldX energies.**

A   Flow chart of the analysis performed in this work.

B   Classification of missense mutations. Class 1a mutants can impair domain-domain inhibitory interactions, as found in the phosphatase PTPN11 or the GEF Sos1. Class 1b mutants may release an inhibitory protein segment, thus resulting in activation, for example mutations in activation segment in kinases, such as BRAF. Class 2 mutants affect protein-protein interactions and can result in a gain in signalling results from the loss of interactions with inhibitors or deactivating proteins. Examples are RAS mutations that prevent the down-regulation by RASA1, or the binding of 14-3-3 proteins, which has been shown to interfere with Ras binding and inhibit Ras-mediated plasma membrane recruitment of RAF1. Class 3 mutants destabilise a globular domain preventing folding. Class 4 mutants can affect protein localisation and/or half-life. Class 5 mutants are neutral and are often localised on the protein surface.

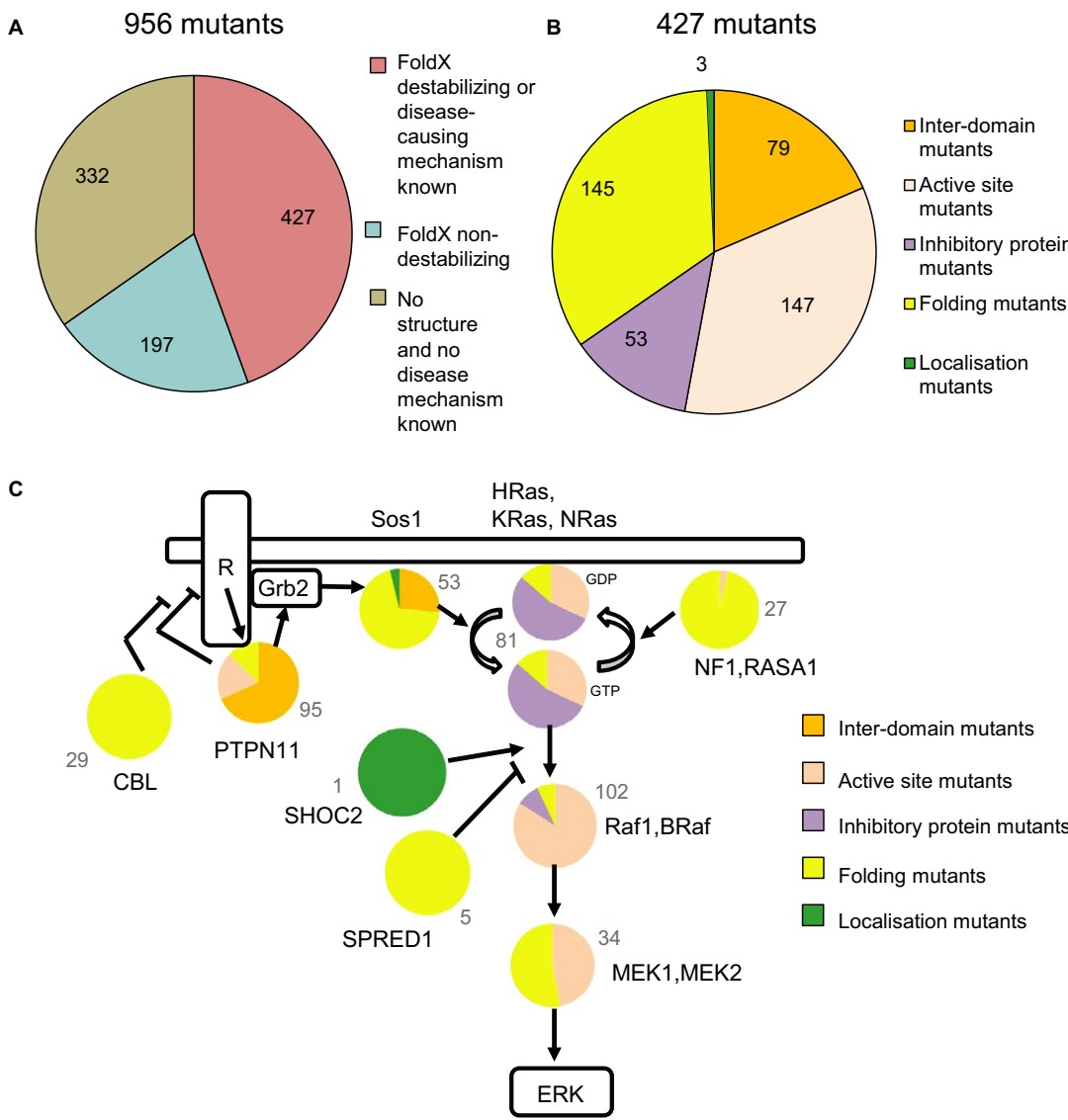

**Figure 4.  Summary of the results from structure-based energy calculations using FoldX.**

A   Structural coverage and the number of mutations predicted to be destabilising and non-destabilising by FoldX.
B   Distribution of 427 missense mutations that map to a 3D structure into different classes.
C   Summary of the predicted effect for the 14 disease genes (proteins represented as pie diagrams). Proteins are arranged in a network similar as in Fig 1A. The grey numbers indicate the total number of different mutations represented in the pie diagrams.

well  (*P*-value = 0.017  by  *t*-test),  especially  at  extreme  values (Fig 5).

   In summary, our FoldX-based analysis supports the hypothesis that differences between the two types of diseases (RASopathy and cancer) manifest through the magnitude of their energy changes, which in many cases will result in altered activities and thus influence the strength of signalling activity.

### Network modelling of mutants with composite and partly compensatory effects

Previous experimental observations identified composite and partly compensatory effects of Ras mutants with respect to binding to

GEF and GAP, and effectors (Gremer *et al*, 2011; Smith *et al*, 2013). Here, we also predicted for some Ras mutants that in addition to preventing Ras GTP hydrolysis (GAP), they also decrease binding to the Ras-activating GEF protein (Supplementary Fig S12). Thus, as GEF is activating, and GAP deactivating, the net result on Ras activation is predicted to be partly compensatory. To analyse the effect of composite and partly compensatory mutations on the network outcome, we constructed a simple computational model of Ras activation, deactivation and effector binding (Fig 6A, Supplementary Table S3). In addition to the GEF- and GAP-catalysed nucleotide exchange rates, we also included intrinsic GDP nucleotide exchange and intrinsic GTP hydrolysis rates (Gremer *et al*, 2011). As some of the Ras mutants have also impaired effector

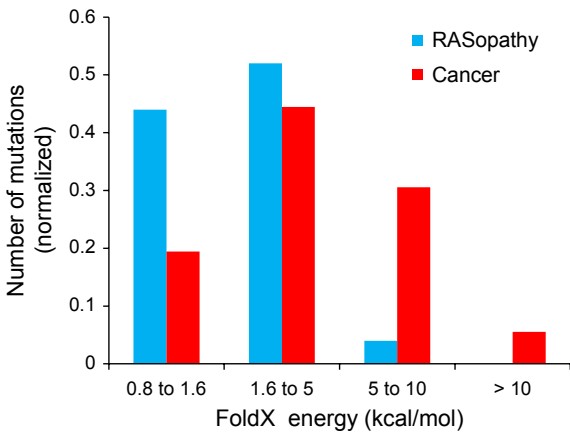

**Figure 5.  FoldX energy changes for a gold set of RASopathy and cancer mutants.**

binding (Gremer *et al*, 2011), we included active Ras binding to effectors, such as RAF1 and BRAF. In this computational model, all five rate constants affecting catalytic activities of Ras or binding to Ras ($k_{GEF}$, $k_{GAP}$, $k_{GDP\ exch.\ intr.}$, $k_{GTP\ hydr.\ intr.}$ and $k_{off}$) are available from *in vitro* biophysical experiments for Ras WT, Ras G12V (cancer) and several RASopathy mutations (K5N, V14I, Q22E, Q22R, P34L, P34R, T58I, G60R, E153V, F156L) (Gremer *et al*, 2011) (Supplementary Tables S3–S6, Materials and Methods). We simulated Ras WT, Ras G12V and all Ras RASopathy mutations, in each case substituting the five rate constants by the experimentally measured rate constants and analysed the binding of active Ras to the effector in equilibrium as the network outcome (RasT_EFF) (Fig 6B). Protein abundances were averages from previous measurements in three mammalian cell lines (Kiel *et al*, 2014) and were kept constant (as wild-type) for all mutant network simulations (Supplementary Table S6). Interestingly, all (except two) Ras RASopathy mutants show equilibrium complex formation abundances (RasT_EFF) that are intermediates between the ones of Ras WT and Ras G12V (Fig 6B). Thus, when detailed experimental data for binding and catalysis are known, the distinction between RASopathies and cancer mutations improves.

We also find an good overall agreement between *in vitro* measured rate constants (Gremer *et al*, 2011) and FoldX energy values, especially for the destabilisation of mutants at the active site and intrinsic and GAP-catalysed exchange reactions (correlation coefficients of 0.65 and 0.43, respectively; Supplementary Table S7). This opens the possibility to estimate rate constants for Ras mutants that have not been measured experimentally based on structure-energy calculations and to integrate these into computational network models.

## Discussion

RASopathies are a class of disorders with overlapping disease symptoms that arise from mutations in different genes (locos heterogeneity). Here, we describe for each of the 15 RASopathy-related proteins their main function within the Ras-MAPK pathway, the location of disease mutations, the energetic effects calculated by

FoldX and integrate this with available functional and biochemical information from the literature.

We found that most of the 15 proteins associated to disorders are expressed in most of the tissues, but at different levels (Su *et al*, 2002; Geiger *et al*, 2013). Variable expression levels in tissues (particularly in stem cells and progenitor populations) have been previously proposed to explain some non-redundant functions for specific Ras isoforms (Schubbert *et al*, 2007a). We show that RASopathy proteins participate in many of the 33 signalling pathways that regulate growth and differentiation, which could explain the overlap in the disease symptoms. This is in support of the 'hour-glass model' of singling, where a small number of pathway components are connected to a large number of receptors (Citri & Yarden, 2006); however, we cannot exclude that some bias may exist as RAS/MAPK proteins are well studied and thus may reported more frequently in pathway databases. Interestingly, RASopathy proteins are excluded from early embryonic signalling pathways, suggesting that those pathways are less plastic and do not tolerate even minor increases or decreases in signalling flows along those pathways.

It is important to understand on the molecular level, why some of the mutations (involving the same protein) can give rise to a drastic phenotype, such as cancer, but others lead to milder effects, as seen in the RASopathies (Keilhack *et al*, 2005). Network-level insights into genetic disorder came from the distinction between two different modes of network changes as the mechanism of underlying phenotypic changes (Zhong *et al*, 2009): a protein removal (as a consequence of a truncation or strongly destabilising mutation) should lead to the loss of all interaction (edges) partners, while a missense mutation on the surface of a protein could affect only few out of all interaction partners ('edgetic perturbation' model). While the enrichment of mutations causing different disorders in different Pfam domains has been experimentally proven for some cases (Zhong *et al*, 2009; Wang *et al*, 2012), we find here that RASopathy and cancer mutations in many cases cannot be distinguished based on the localisation in different domains, or in many cases specific positions. Our structural analysis and FoldX energy calculations show that most of the missense mutations with high destabilisation energies are predicted to affect protein activation or inhibition by affecting autoinhibitory domain-domain or domain-protein segment interactions, or through a loss of binding of inhibitory proteins. Only in the case of proteins that reduce activity of critical nodes, we find a significant number of mutations that could compromise protein folding (i.e. for RASA1 and NF1). Finally, we found also few mutations that could affect protein localisation (i.e. RASA1 and SOS1). For a few mutations on the surface, we could not see any change in stability, and based on sequence conservation, we postulated that they could be involved in protein-protein interactions which can be tested experimentally. Thus, a change in activity seems to be the most prevalent disease-causing mechanism. However, through our analysis, we cannot exclude that changes in interaction partners could additionally play a role. For example, BRAF 600E mutations treated with a kinase inhibitor can still activate the MEK-ERK pathway through heterodimerisation with RAF1 (Heidorn *et al*, 2010).

Energy calculations also revealed that activating somatic mutations in general have higher energies compared to the germline mutations. This thereby supports the hypothesis that the difference

   

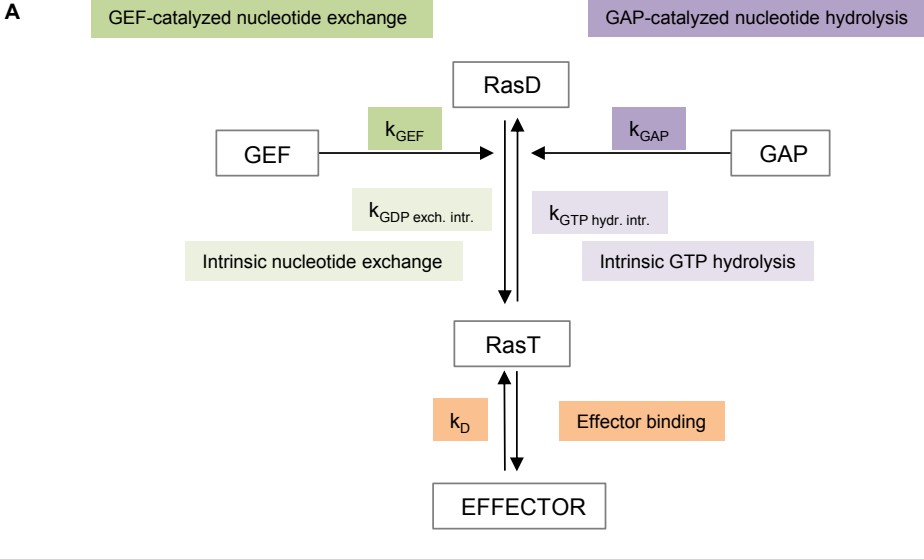

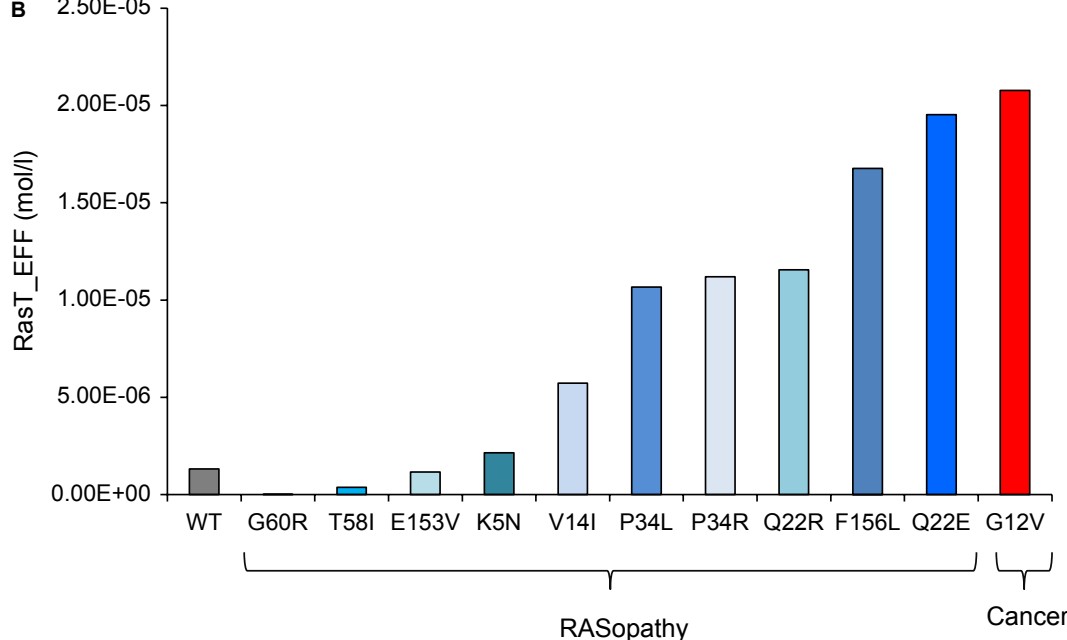

**Figure 6.   Schematic network model representation and model simulation results.**

A   Schematic representation of the network model. The 5 indicated rate constants represent the five experimentally measured rate constants for Ras WT and mutants. $k_{GEF}$ is the GEF catalysed GDP dissociation, $k_{GAP}$ is the GAP-catalysed GTP hydrolysis, $k_{GDP\ exch.\ intr.}$ is the intrinsic nucleotide dissociation, $k_{GTP\ hydr.\ intr.}$ is the intrinsic GTP hydrolysis, and $k_D$ is the affinity between Ras binding to the effector (EFF).

B   Results of the amount of Ras binding to effectors (Ras_EFF) in equilibrium for 12 network model simulations for using the respective 5 rate constants for Ras WT, Ras G12V (cancer) and RASopathy mutations (Ras K5N, V14I, Q22E, Q22R, P34L, P34R, T58I, G60R, E153V, F156L).

between cancer and RASopathy mutations lay in part in their quantitative effect on signalling, which emphasises the importance of a quantitative systems analysis of signalling networks. In fact, the degree and duration of Ras-MAPK activation can have profound effects on cell fate decision (e.g. PC12 cells) (Marshall, 1995). Also, this explains why a cancer mutation might often be embryonic lethal, as shown for the BRAF V600E mutation (Mercer *et al*, 2005). It is, however, true that the energy distributions of cancer and

RASopathy mutations largely overlap, which probably precludes general diagnostic implications such as the prediction about whether RASopathy patients develop cancer. This of course could be due to the fact that we are grouping mutations in different positions and proteins to have enough statistical power. This blurs specific position effects that should be considered to have diagnostic power. We have demonstrated for phenylketonuria and retinitis pigmentosa diseases that, when working with well-characterised

individual proteins and experimental biochemical data on binding or catalysis are available, FoldX-based quantitative stability prediction correlate well with the onset of the disease (Pey *et al*, 2007; Rakoczy *et al*, 2011). In fact, we showed here that when focusing on a single protein (i.e. Ras) and using a toy network model in which experimental data for binding and catalysis were introduced (Gremer *et al*, 2011) effects, the distinction between RASopathy and cancer mutations can be improved. Thus, to have true predictive power, detailed analysis at each individual position should be performed and different factors considered. For example, it has also been proposed that Raf-1 could increase the intrinsic GTP hydrolysis on Ras, which has been associated with different transforming activities of mutations at position Q61 of Ras (Buhrman *et al*, 2007). Other effects, such as the preferential expression of isoforms during development or even different localisation dynamics (Chandra *et al*, 2012), should be taken into account to explain why Costello syndrome mutations that harbour a G12X mutation in HRAS do not show up frequently in cancer (although Costello syndrome patients do develop tumours more frequently; Gripp & Lin, 2012), while KRAS G12X mutations are frequently involved in cancer.

In conclusion, through our study, we have extended the knowledge on the disease-causing mechanism of RASopathies. As a central outcome of this work, we suggest that quantitative changes in overall activity of the pathway, more so than rewiring or perturbation of specific interactions could explain the difference between RASopathy and cancer mutations. Our work shows that even for the same protein, depending on which disease mutation is affected, the effect on the network will be different, ranging from changes in abundance to changes in rate constants and affinities. This is a relevant finding for future system approaches that aim to combine tissue- or patient-specific protein abundances with disease networks in a quantitative way: in addition to protein abundances (node sizes in a network) and the presence and absence of interactions (qualitative edges or 'edgetics' in a network), quantitative effects on edges through changes in affinities and kinetic constants need to be considered (quantitative edges or 'enedgetics'). Structure-energy predictions are a crucial step in dissecting the different cases and enable us to calculate estimates for these rate constants that can then be integrated into predictive mathematical models of disease networks.

# Materials and Methods

## Mutation databases and protein sequence analysis online tools

Germline mutations for the 15 RASopathies-associated genes were extracted from OMIM (http://www.omim.org/) and Uniprot (http://www.uniprot.org/). Somatic mutations were collected from COSMIC (http://cancer.sanger.ac.uk/cancergenome/projects/cosmic/) and cBioPortal (http://www.cbioportal.org/public-portal/) (Supplementary Table S1). Disease phenotypes were retrieved from OMIM (http://www.omim.org/). Shannon entropies were calculated using PVS (Protein Variability Server, http://imed.med.ucm.es/PVS/) using a multiple sequence alignment generated using ClustalW (http://www.ebi.ac.uk/Tools/msa/clustalw2/) for protein sequences from different organisms (retrieved from the

HomoloGene database at http://www.ncbi.nlm.nih.gov/homologene).

## Domains and three-dimensional protein structures

Protein domains were predicted using Pfam (http://pfam.sanger.ac.uk/). Protein structures were retrieved from the protein data bank (http://www.rcsb.org/pdb/home/home.do) (Supplementary Table S1). For SHOC2, the leucine-rich repeats (LRR) were assigned based on a recent homology model (Jeoung *et al*, 2013). As no crystal structure is available, we only provide the mapping of the disease mutation to the different LRR repeats, and based on the LRR alignment, we indicate mutations of conserved hydrophobic residues that may affect the protein stability. No crystal structure is available for Rit1. As sequence homology outside the effector binding region is low, it was not possible to build reliable homology model using FoldX. We nevertheless indicate the corresponding residues in Ras for each mutation in Rit1.

### Domains and structural coverage of PTPN11
The N-terminal SH2 domain interacts intra-molecularly with the PTP domain and thereby inhibits catalytic activity and access of the substrate to the catalytic site (Barford & Neel, 1998; Hof *et al*, 1998). The crystal structure of (nearly) full-length PTPN11 has been solved (pdb entry 2SHP) (Hof *et al*, 1998).

### Domains and structural coverage of SOS1
SOS1 contains an N-terminal histone domain, followed by a pleckstrin homology (PH) domain, the RAS exchanger motif (REM) and the catalytic Cdc25 domain. The C-terminus contains proline-rich regions for the recognition of SH3-containing upstream adaptor proteins, such as Grb2. The Ras GEF activity of the Cdc25 domain is controlled by intra-molecular interactions of the DH and Rem domains, which stabilise SOS1 in its inactive conformation (Sondermann *et al*, 2004). Membrane recruitment promotes conformational changes that turn on the GEF activity and also unmask a distal binding site for RAS (GTP-bound) that is otherwise occupied by the DH domain (Margarit *et al*, 2003). The crystal structures of SOS1 spanning the DH, PH, Rem and Cdc25 domains (pdb entry 1XD4) (Sondermann *et al*, 2004), the Rem and Cdc42 domain in complex with nucleotide free Ras and Ras (nucleotide-bound) at the distal binding sites (pdb entry 1XD2) (Sondermann *et al*, 2004), the DH and PH domains (pdb entry 1DBH) (Soisson *et al*, 1998) and the histone domain (pdb entry 1Q9C) (Sondermann *et al*, 2003) have been solved.

### Domains and structural coverage of RASA1 and NF1
RASA1 gets recruited to membrane receptors such as ErbB family members through binding of its SH2 domain to the phosphorylated receptor (Jones *et al*, 2006). The crystal structure of the SH2 domain of RASA1 has been solved (pdb entry: 2GSB). The complex structure of the SH2 domain of Nck1 with a phospho-peptide (Frese *et al*, 2006) was used to analyse whether disease mutations are in the peptide binding area. For NF1, the structures of the GAP domain (pdb entry 1NF1) (Scheffzek *et al*, 1998) and the Sec14-PH domains (pdb entry: 3PG7) (Welti *et al*, 2011) have been solved. The Sec14 and PH-like domain follows the GAP domain, and together they are termed GAP-related domain (GRD).

### Domains and structural coverage of RAS

The crystal structure of Ras has been solved in complex with the GAP domain of RASA1 (pdb entry 1WQ1) (Scheffzek *et al*, 1997), the Ras-binding domain (RBD) of RAF1 (pdb entry 1GUA) (Nassar *et al*, 1996), and in complex with SOS1 (see before, pdb entry 1XD2) (Sondermann *et al*, 2004).

### Domains and structural coverage of RAF1, BRAF, MAP2K1 and MAP2K2

The crystal structure of Ras has been solved in complex with the Ras-binding domain (RBD) of RAF1 (pdb entry 1GUA) (Nassar *et al*, 1996), and the complex between 14-3-3 and a peptide spanning S259 of RAF1 (pdb entry 3IQJ) (Molzan *et al*, 2010). A complex structure of the kinase domains of RAF1 and BRAF has been solved (pdb entry 3OMV). Furthermore, the kinase domains of MAP2K1 and MAP2K2 have been solved (pdb entries 2Y41 and 1S9I), and the complex between MAP2K1 and KSR1 (pdb entry 2Y41).

### Domains and structural coverage of SPRED1

The crystal structure of the WH1 domain of SPRED1 has been solved (pdb entry 3SYX).

### Domains and structural coverage of CBL

The crystal structure of the trimeric complex of the Cbl_N, Cbl_N2, Cbl_N3 and zf_C3HC4 domains of CBL in complex with the E2 domain of UbcH7 and a peptide of the ZAP-70 receptor (pdb entry 1FBV) (Zheng *et al*, 2000).

## Protein mutations and stability predictions by FoldX

FoldX (http://foldx.crg.es/) is a computer algorithm that allows the calculation of interaction energies contributing to the stability of proteins and protein complexes (Guerois *et al*, 2002; Schymko-witz *et al*, 2005). For details concerning the force field, please see the description in the online version and in related publications. The FoldX algorithm allows predictions of mutational affect for any of the 20 natural amino acids, but it does not allow any back-bone changes. Thus, for example, predictions for RasQ61 mutations may not be reliable, as the structure of the Ras Q61L mutation has been solved and it shows larger structural changes (Buhrman *et al*, 2011). This is especially important for mutations in loops (e.g. when modelling Ras G12V in complex with GAP); in this case, introducing mutations results in van der Waals clashes and therefore large energy changes, which, however, will not unfold the protein. Prior to any mutagenesis, the RepairPDB option of FoldX was used to optimise the total energy of the protein, by identifying and repairing those residues that have bad torsion angles and van der Waals clashes. Mutagenesis was performed using the BuildModel option of FoldX. The stabilities were calculated using the Stability command of FoldX, and $\Delta\Delta G$ values are computed by subtracting the energy of the WT from that of the mutant.

## The FoldX energy function

The FoldX energy function includes terms that have been found to be important for protein stability. The free energy of unfolding ($\Delta G$) of a target protein is calculated using equation:

$$\Delta G = Wvdw * \Delta Gvdw + WsolvH * \Delta GsolvH + WsolvP * \Delta GsolvP$$
$$+ \Delta Gwb + \Delta Ghbond + \Delta Gel + \Delta GKon + Wmc * T * \Delta Smc$$
$$+ Wsc * T * \Delta Ssc$$

with:

- $\Delta Gvdw$ as the sum of the van der Waals contributions of all atoms with respect to the same interactions with the solvent.
- $\Delta GsolvH$ and $\Delta GsolvP$ as the differences in solvation energy for apolar and polar groups, respectively, when these change from the unfolded to the folded state.
- $\Delta Ghbond$ as the free energy difference between the formation of an intra-molecular hydrogen bond and intermolecular hydrogen bond.
- $\Delta Gwb$ as the extra stabilising free energy provided by a water molecule making more than one hydrogen bond to the protein (water bridges) that cannot be taken into account with non-explicit solvent approximations.
- $\Delta Gel$ as the electrostatic contribution of charged groups, including the helix dipole.
- $T * \Delta Ssc$ as the entropy cost of fixing the backbone in the folded state.
- $\Delta Ssc$ as the entropic cost of fixing a side chain in a particular conformation.

If interaction energies between protein complexes are calculated, two additional terms are needed:

- $\Delta GKon$, which reflects the effect of electrostatic interactions on the association constant kon (this applies only to the subunit binding energies)
- $\Delta Str$, which is the loss of translational and rotational entropy that ensues on formation of the complex. The latter term cancels out when we are looking at the effect of point mutations on complexes.

## Network simulations

A simplified mathematical model involving Ras activation, deactivation and effector binding node, was constructed based on mass action kinetics using the iNA simulation software (Thomas *et al*, 2012). The model includes GEF- and GAP-catalysed nucleotide exchange rates, intrinsic GDP nucleotide exchange and intrinsic GTP hydrolysis, and active Ras binding to effector molecules (EFF), such as RAF1 and BRAF (Fig 6A). The initial concentrations of species in the model (i.e. total Ras [=sum of H, K, and N-Ras], GEF [=Sos1], GAP [=RASA1] and EFF [=sum of RAF1 and BRAF]) were averages based on experimentally determined protein abundances in three mammalian cell lines (Kiel *et al*, 2014) (see Supplementary Table S5). The rate constants for GEF binding and catalysis, GAP binding and catalysis, and effector binding were taken from a previous model (Kiel & Serrano, 2009) (see Supplementary Tables S3-S6). The off-rates for binding of GAP and GEF to Ras were decreased 10-fold in order to account for that those enzymes should already be localised at the membrane (e.g. through recruitment by receptor/scaffold interactions). The intrinsic rate constants of GTP hydrolysis and nucleotide exchange were taken from experimental measurements (Gremer *et al*, 2011). The considered reaction volume was 1e-14 litres. Ras was initially considered to be GDP-bound. GTP or GDP binding to Ras was not modelled explicitly. The assumption is that free nucleotide is highly abundant in the cell and Ras is always

bound to nucleotides. This is comparable to modelling kinase activities, where ATP binding is not explicitly modelled. Binding of active Ras to the effector was analysed in equilibrium conditions as a result of the network modelling in equilibrium. For modelling Ras G12V (cancer) and several RASopathy mutations (K5N, V14I, Q22E, Q22R, P34L, P34R, T58I, G60R, E153V, F156L) for each mutation, the rate constants were changed according to experimental measurements performed for intrinsic GTP hydrolysis and nucleotide exchange, the GAP and GEF stimulated, and effector binding (Supplementary Table S5). As GAP and GEF-reactions measured by Gremer $et$ $al$, 2011 represent present $k_{obs}$ rates and thus depend on the actual concentration of enzyme used in the experiment, we calculated fold changes compared to wild-type for kobs rates and translated those fold changes into the rate constants used in previous models (Kiel & Serrano, 2009; Supplementary Table S5). We are not considering a wild-type allele in our simulations.

**Supplementary information** for this article is available online: http://msb.embopress.org

## Acknowledgements

We thank Marius Ueffing for valuable discussions on the manuscript. This work was supported by the EU (PRIMES under grant agreement number FP7-HEALTH-F4-2011-278568). This work was supported by the Spanish Ministerio de Economía y Competitividad, Plan Nacional BIO2012-39754 and the European Fund for Regional Development. We acknowledge support of the Spanish Ministry of Economy and Competitiveness, 'Centro de Excelencia Severo Ochoa 2013-2017' (SEV-2012-0208).

## Author contributions

CK and LS conceived and designed the study. CK performed the computational analyses. CK and LS wrote the manuscript.

## Conflict of interest

The authors declare that they have no conflict of interest.

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
