## [Review Process File · Molecular Systems Biology]

Structure-energy-based predictions and network modeling of RASopathy and cancer missense mutations

Christina Kiel, Luis Serrano

Corresponding author: Christina Kiel, Centre for Genomic Regulation CRG

Review timeline:

Submission date:	02 January 2014
Editorial Decision:	06 February 2014
Revision received:	06 March 2014
Accepted:	25 March 2014

Editor: Maria Polychronidou

Transaction Report:

1st Editorial Decision

06 February 2014

Thank you again for submitting your work to Molecular Systems Biology. We have now heard back from the three referees who agreed to evaluate your manuscript. As you will see from the reports below, the referees acknowledge that the presented analysis of RASopathy mutations in relation to cancer mutations yields potentially interesting findings. However, they raise a series of concerns, which should be carefully addressed in a revision of the manuscript.

Without repeating all the points listed below, most of the reviewers' comments refer to the need to clarify and better document several points throughout the manuscript. In particular, the model describing Ras activation and the related analyses should be better explained and the potential issues arising from the difficulty in distinguishing between RASopathy and cancer mutations (i.e. due to the fact that RASopathy patients sometimes develop cancer) should be discussed.

 Reviewer #1:

Kiel and Serrano address the question of what distinguishes RASopathy mutations from cancer-associated mutations in the RAS/ERK pathway. The authors assemble a large number of missense mutations from various databases and use FoldX to assess energetic consequences of individual mutations relative to WT. The main claim is that quantitative, rather than qualitative differences distinguish the mutations. Specifically, they find that mutations related to cancer have higher destabilization energies compared with those found in RASopathies, although those mutations distribute equally over domains and interfaces. As a secondary point, they also suggest that compensatory effects on activation/inactivation events in individual signaling components might also contribute to the weaker effects of RASopathy mutations.

Although there are some interesting findings in this manuscript, the concept that RASopathies differ quantitatively from cancer mutations was proposed at the outset of the RASopathy field, and has already been subjected to substantial experimental support (much of which was not referenced by the authors). Specifically, the Neel lab has provided substantial biochemical (Keilhack et al, 2005) and biological (Araki et al, 2009) evidence that the extent of SHP2 activation determines the effects of PTPN11 mutations, whereas Marais and Pritchard recently provided similar evidence for BRAF mutations associated with CFCS and melanoma, respectively, using mouse models. The present manuscript makes a valuable contribution by extending this to a much larger number of mutations, but it is unclear that it reaches any novel conclusions. Moreover, it makes broad, sweeping conclusions based on what is, at times, highly selective consideration of the data.

Specific Points:

- 1) The authors claim that RAS/MAPK proteins are involved in more pathways than others, and are widely expressed. Although this might well be true, the first conclusion suffers from selection bias: obviously, as an extremely well-studied signaling module, it is, all else being equal, likely to be reported to be involved in more pathways than others. Also, the latter is virtually tautological: components of this pathway belong to the same module, so as long as one is expressed, it is likely they all will be.
- 2) The authors conclude that RASopathy mutations have lower destabilizing energy values than do cancer mutations. Although this conclusion is likely to be true, the analysis selectively disregards some mutations, eventually using only a 'gold set' that had already been studied experimentally. Even then, whereas the destabilizing energy of cancer mutations as a class is clearly greater in Fig. 5, there is a considerable overlap between the two types of mutations. Hence, the utility of their conclusion is questionable: in other words, it is not clear that one could examine a new mutation by FoldX and classify it as likely cancer-causing or not. If the authors could, in fact do this (and demonstrate it convincingly), their findings would be of considerable interest for counseling and monitoring RASopathy patients about their cancer risk. However, without such ability, the claim that their approach has therapeutic or diagnostic implications does not seem warranted (and they certainly don't provide much discussion as to how/why it would be)
- 3) This model needs to be more clearly explained in the text. What are the starting conditions being modeled? The system starts with Ras-GDP, but what is the concentration of free GTP? After exchange followed by hydrolysis Ras returns to the GDP-bound state. Was the free GTP all hydrolyzed or is this the steady-state in the presence of GTP? The figure legend for panel C is exactly the same as B, although it appears to measure effector activation rather than Ras-GTP.

Minor concern:

"RAS/MAPK proteins are involved in more pathways than other proteins": While this is probably true, this trend must also reflect the fact that these 12 proteins are disproportionately well studied with more publications in the literature to implicate them in more pathways.

- 4) The authors state: "We predicted for some Ras mutants that in addition to preventing Ras GTP hydrolysis, they also decrease binding to the Ras activating GEF protein. More interestingly, the compensatory energies are stronger for RASopathies compared to cancer mutations, which could explain the mild phenotype of RASopathy compared to cancer mutations. " Yet by comparing the ratios for FoldX values for RasGAP versus RasGEF in cancer versus RASopathy, it is not clear to me that there is any difference. GEF interactions are more destabilized by RASopathy mutations than are cancer mutations, but so are GAP interactions. Has a statistical analysis of the GAP/GEF energy ratios been performed?
- 5) The model in Fig. 7 should be more clearly explained in the text. For example, what are the starting conditions? The system starts with RAS-GDP, but what is the concentration of free GTP? After exchange, followed by hydrolysis, RAS returns to the GDP-bound state. Was the free GTP all hydrolyzed or is this the steady-state in the presence of GTP? Also, the legend for panel C is exactly the same as B, although it appears to measure effector activation rather than RAS-GTP.
- 6) It is unclear why the authors include RASA1 as a RASopathy gene; most would not. Also unclear is why they fail to consider RIT1, which was reported to be a major remaining gene for NS last year

by Aoki et al.

7) Why don't the authors include colon cancer as a disease in which KRAS mutations are common (~50% of cases)?

8) Similarly, it is not clear why the authors state that MEK mutations have not been observed in cancer. In fact, N-terminal MEK mutations have been reported in lung cancer and as resistance mutations in melanoma.

9) Many of the references are incorrect and/or inappropriate. Some times primary papers are cited that do not have the data stated, other times, reviews are cited that are not always up to date. Overall, the referencing is quite selective.

Reviewer #2:

Summary

In this manuscript the authors aim to explain the differences between the mutations in the Ras/MAPK genes that lead to the development of cancer or RASopathies. Interestingly, different mutations in the same gene, and in some cases even in the same residue, result in different outcomes. Thus, the authors try to shed light on an intriguing and relevant question that is difficult to address by using only experimental approaches. This manuscript by Kiel & Serrano analyses 508 different missense mutations in the 12 genes related to RASopathies and cancer using a computational approach. By combining protein network data with structure-energy based predictions the authors conclude that quantitative rather than qualitative differences in Ras dependent signalling pathway determine the phenotype. The authors also develop a simple mathematical model to simulate the effects of some of the different mutations in the Ras/MAPK network to further support their conclusions. The authors conclude that the systemic approach used in the current study that integrates structure-energies predictions and predictive mathematical modelling is a valid method to study the different effects on a signalling network cause by different mutation in the same proteins

General remarks.

The work described by Kiel and Serrano is in general well supported by the data presented. The systematic computational analysis described in this study can be applied to other signalling networks. It also demonstrates the importance of considering protein structure in order to get a better understanding of the molecular mechanisms how signalling networks work.

Importantly, the data presented here helps to understand the different role of Ras/MAPK mutants in cancer and RASopathies. Thus, this manuscript extends our knowledge on disease-causing mechanisms of RASopathies. The authors present a thorough analysis of the mutations in the 12 genes responsible using a combination of known data curated from the literature and several databases with new computational approaches. They are able to explain different phenotypes observed for different mutations through network analysis and protein structure. Their conclusions from these studies are supported by good correlation with experimental data from the literature (Table S6) and predictions of the mathematical model generated. It must be noted that this computational approach allows us to get insight into the biological processes without the need to perform additional experiments. It is, however, dependent on the availability of protein structures (in this case 11 out of 12, but some relevant regions are missing).

This work can be used to estimate effects of new, uncharacterised mutations. The manuscript is interesting for a wide audience including structural and computational biologist but also researches interested in signal transduction. Overall this manuscript is relevant for the audience of this journal.

Specific comments and suggestions through manuscript.

Major points.

Although the manuscript fulfils the criteria for its publication on MolSysBiol this reviewer thinks that some editing is necessary before its publication. There are numerous semantic and grammatical mistakes in the text (see below), and the text will benefit from rewriting in a more concise and careful way. The introduction is rather long and some of the points made are not necessary. Specifically, the

description of some of the results in this part can be shortened since they are explained in the results and discussion. Also, I would suggest the author to consider editing the first paragraph of the classification of disease mutations and structural coverage. It reads as introduction within the results section and may be better to be included in the actual introduction.

I also would suggest moving Figures S5 and S6 into the main text. These figures are describing the domain analysis of missense mutations in RASopathies and cancer, which would deserve to be shown in the main text rather than as supporting material.

The oncogenic RasG12V mutation is known to lock Ras in an active conformation caused by the loss of GTPase activity. However, in the mathematical model predictions RasV12T is shown to come back to an inactive state. How do the authors explain this? Activation of wt Ras proteins has been shown to be important for oncogenic Ras transformation (Jeng HH et al. 2012, Nature Communications; Young A, 2013, Cancer Discovery; Matallanas et al. Molecular Cell, 2010). Does the model consider total Ras activation including the other Ras isoforms and/or the wt allele of the mutated isoform? Is this what is shown in this prediction?

The Ras Q61 mutant is mentioned the first time in the discussion section, why is it not used in the analysis or as prediction?

Minor points

Page 4: SPRED1 prevents RAF1 activation by Ras, but is NOT an inhibitor of the Ras-RAF1 interaction.

Page 8: Reference Goh et al. requires re-formatting

Table S1: Formatting in the download is very poor. Please make sure that this corrected in the final version.

Page 9: Why did the authors choose this database over others? Wrong reference: Kandasamy, K. and Mohan, SS. et al. (2010) NetPath: A public resource of curated signal transduction pathways. Genome Biology. 11:R3

Reference in figures S3 and S4 are missing (Geiger et al 2013, Su et al 2004) and this reviewer failed to find them with the information provided in PubMed.

Page 11: SPRED1 is used here as an example but the authors say earlier (fig 1B) that it has no cancer-related mutations. Please clarify.

Page 11: "An exception is SPRED1, but here the total number of mutations is low and probably not statistically significant." What does "probably not statistically significant" mean? Statistical significance can be tested and a result can be significant or not.

Page 11: "For MAP2K1 and MAK2K2 no cancer mutations are identified." This is incorrect. See Cancer Res. 2008 Jul 15;68(14):5524-8. doi: 10.1158/0008-5472.CAN-08-0099. Novel MEK1 mutation identified by mutational analysis of epidermal growth factor receptor signaling pathway genes in lung adenocarcinoma.; Marks JL, Gong Y, Chitale D, Golas B, McLellan MD, Kasai Y, Ding L, Mardis ER, Wilson RK, Solit D, Levine R, Michel K, Thomas RK, Rusch VW, Ladanyi M, Pao W. Cancer Res. 2011 Aug 15;71(16):5535-45. doi: 10.1158/0008-5472.CAN-10-4351. Epub 2011 Jun 24. Identification of the MEK1(F129L) activating mutation as a potential mechanism of acquired resistance to MEK inhibition in human cancers carrying the B-RafV600E mutation. Wang H, Daouti S, Li WH, Wen Y, Rizzo C, Higgins B, Packman K, Rosen N, Boylan JF, Heimbrosk D, Niu H.

This reviewer would suggest to re-format figure 3 by splitting it into parts A and B, as the second part explains the different classes and the first part the procedure (and is not referred to in text). The structures in the second part are referred to as pdb-id in the figure while their names are used in corresponding text. Please relate them to each other.

Page 16: Figure 6D+E are merged in text and legend

Page 17: Legend for figure 7C is wrong (identical to 7B)

Please correct the numerous grammatical errors and wrong use of words, e.g.:

Page 4: "RASA1 encodes for the p120..."

Page 4: "Binding of 14-3-3 proteins have been shown..."

Legend to Figure 2A "...the colour and size of the nodes represents the number of signalling pathway they predicate in (see legend)." - this sentence contains two errors, i.e. represents instead of represent, and predicate instead of participate.

Page 16: "...is only minor increased..."

Page 19: "We show that the majority of the mutations involved in the twelve RASopathy proteins that we could model and resulted in significant energy changes (73%)."

Page 20/21: BRAF 600E is not kinase impaired. Beside, mutations in BRAF and KRAS are usually mutually exclusive. Please correct.

The authors state that "with the exception of H-Ras and MAP2K2, all RASopathy genes are embryonic lethal when knocked out [in mice]". This is not true. N-Ras and K-Ras4A mice are viable and only K-Ras4B is embryonic lethal (Plowman et al 2003, and Umanoff 1995). Please correct. The authors state that "RAF1 while producing the same RASopathies, it is not involved in tumour formation". This is not true. Although rare, mutations on this gene have been described in the literature (Zebisch et al 2006). Furthermore, the involvement of this protein in cancer is also suggested by the re-activation of ERK in patients with mutant B-Raf treated with vemurafenib which seems depends on Raf-1. Please correct and clarify what you mean.

There is inconsistency in the nomenclature of the proteins i.e. both Raf1 and CRaf are used.

Reviewer #3:

The paper by Kiel and Serrano is a systems biology approach in order to better understand RASopathy mutations and their relation to cancer mutation. RASopathies like Noonan, Leopard or CFC syndrome are developmental diseases mutated in genes that can be aligned in a single pathway that goes from activated RTKs downstream to activation of Erk and involves mainly Ras and its regulators and effectors. What makes such a study particularly interesting is that the same genes are also mutated in cancer which begs the question why these mutations don't lead to cancer (which some do, see below). The authors first collect and order the large number of patient mutations and make a wonderful representation and categorization of the diseases. They show that these proteins participate in early to late developmental pathways which is why such patients live. They observe that RASopathy and cancer mutations cannot be distinguished on the bases of the type of domains which are mutated. They then use the program FoldX to calculate the change in energy for those proteins and/or domains for which a three-dimensional structure is available, the others are analyzed according to the degree of conservation and the energy change expected for such a mutation (be it on the surface). By FoldX and structural considerations (where structures are available) the mutations are then categorized by the type of effect they might have on activity, interaction with regulators and effectors and on folding. Of the 339 mutants, 82% have significant energy changes higher than 0,8 kcal. 93 mutants do not show any significant energy changes and could represent proteins that have a defect in interaction with partner proteins. The authors claim that cancer mutations show larger energy changes than RASopathies.

In a more detailed analysis the authors model the effect of the individual Ras mutations on the Ras-MApK pathway, Relying on a set of biophysical measurements by Gremer et al, they come to the conclusion that Ras activation-inactivation is between the wt and oncogenic G12V Ras, supporting the notion that Ras activation in the diseases is less severe and sustained than in RASopathies.

Overall this is a nice summary and investigation that broadens our overall understanding of the diseases which needs however some explanations, clarifications and corrections.

The separation of mutation into the two categories is not so straightforward, as kids with RASopathies do sometimes develop cancer (i.e. JMML) The data maybe difficult to come by, but should at least be discussed, possibly in terms of severity of the disease.

It would be worth mentioning/discussing Leopard mutations that harbor a G12X mutation in H-Ras,

which do in principle not show as cancer, why?

I also have a slight problem with extrapolating energy changes to changes in activity/affinity ect, which I am certain the authors are aware of, but don't discuss in due form. A good example is G12V (Table S6). They show a severe destabilization by FoldX, where in fact this mutants is quite normal in terms of its biochemical properties, including binding to GAP, except that it cannot be activated because arginine cannot swing into the active site.

My greatest problem is with the modelling of Ras activation, needs more explanation:

What happens at time point zero, and what is the exchange factor reaction that is modeled here, what are the rates being used. The rates experimentally determined were just k_{obs} , for a certain concentration of Ras and regulators and effectors. In the real world, the GEF would be SOS, and SOS is a very weak enzyme that in fact needs to be activated by a feedback (feed forward) mechanism. For the second order reactions one would need concentrations which are given but not explained where they come from (Suppl. Table 3 and 4). They are unlikely to come from expression levels given in the Tables, or do I miss something here.

And why is Ras activation eventually going down to zero? In reality one would expect, at least for the mutants, them to be at a certain level above zero, that's why they cause disease.

Fig7C shows indeed that the activation of effectors is more difficult to model explaining that the model needs some modification. I partially agree with the authors that this might be due to the inability to model Raf activation. This should at least be discussed.

The authors mention mutants G60R which I cant see in Fig. 7b

On p.16, they talk about compensatory mutations , where GTP hydrolysis defect would be compensated by a decrease in SOS binding, which would not be compensatory, and this is in fact mentioned/shown in Fig. 6

The real compensatory mutations have been discovered by Gremer et al., and that should be mentioned on p16, already, in that same chapter.

P19, it is mentioned that Rasopathies arise from mutations in more than one gene, which is misleading, should be rephrased, there is one gene-one disorder in each patient.

P19, we found that most of the..., did the authors do the expression tests???

P21, why should BRAFV600E not activate MEK-ERK?

1st Revision - authors' response

06 March 2014

We have submitted the revised version of manuscript MSB-14-5092, entitled "Structure-energy-based predictions and network modeling of RASopathy and cancer missense mutations".

We like to thank all three reviewers for their positive comments and constructive criticism about the manuscript. We have now addressed all concerns and suggestions of the three referees.

In particular, we explain the Ras activation model better, and for this we have improved the model scheme in fig 7A (now figure 6A). In addition, we have changed the model to an alternative model for Ras activation, where instead of activating Ras through a pulse (by recruiting the GEF to the membrane followed by GEF inactivation), we simulate steady state Ras activation under constant GEF and GAP activities, and effector binding conditions, which possibly better resembles the in vivo conditions of tumour formation. Essentially these simulations results are qualitatively similar. Further, we discuss now potential issues arising from the difficulty of distinguishing RASopathy from cancer mutations, and that RASopathy patients sometimes develop cancer. We also explain better that when working with well characterized proteins and additional experimental biochemical data on binding and catalysis are available, e.g. here for Ras, and using a toy network model in which

experimental data for binding and catalysis are introduced, the distinction between RASopathy and cancer mutations can be improved.

In fact, the most important point of our study is that quantitative information on edges needs to be considered when aiming at generating tissues or patient-specific disease networks.

In addition, we included now 3 more RASopathy genes and updated mutations for the other 12 proteins; the total number of missense mutations analysed is now 956.

Please see the detailed description of the changes made in response to the referees below.

We hope that the new version will be acceptable for publication in Molecular Systems Biology.

Response to reviewer comments:

Reviewer #1:

Kiel and Serrano address the question of what distinguishes RASopathy mutations from cancer-associated mutations in the RAS/ERK pathway. The authors assemble a large number of missense mutations from various databases and use FoldX to assess energetic consequences of individual mutations relative to WT. The main claim is that quantitative, rather than qualitative differences distinguish the mutations. Specifically, they find that mutations related to cancer have higher destabilization energies compared with those found in RASopathies, although those mutations distribute equally over domains and interfaces. As a secondary point, they also suggest that compensatory effects on activation/inactivation events in individual signaling components might also contribute to the weaker effects of RASopathy mutations.

Comment 1a: *Although there are some interesting findings in this manuscript, the concept that RASopathies differ quantitatively from cancer mutations was proposed at the outset of the RASopathy field, and has already been subjected to substantial experimental support (much of which was not referenced by the authors). Specifically, the Neel lab has provided substantial biochemical (Keilhack et al, 2005) and biological (Araki et al, 2009) evidence that the extent of SHP2 activation determines the effects of PTPN11 mutations, whereas Marais and Pritchard recently provided similar evidence for BRAF mutations associated with CFCS and melanoma, respectively, using mouse models.*

Reply 1a: We apologize for this omission. We now mention these papers, as well as others in our introduction.

We have also included three new proteins to the analysis after careful literature search.

Comment 1b: *The present manuscript makes a valuable contribution by extending this to a much larger number of mutations, but it is unclear that it reaches any novel conclusions. Moreover, it makes broad, sweeping conclusions based on what is, at times, highly selective consideration of the data.*

Reply 1b: We have tried throughout the manuscript to make the novel conclusions clearer now:

-Domain and even positions do not discriminate between RASopathy and cancer mutations

-In general cancer mutations result in larger energy changes than RASopathy mutations.

-Even for the same gene, depending on the type of mutation, different disease-causing mechanisms exist. This is a relevant finding for system approaches that aim to combine tissue- or patient-specific protein abundances with disease networks in a quantitative way: in addition to protein abundances and the presence and absence of interactions, quantitative effects on interactions through changes in affinities and kinetic constants need to be considered.

-the importance of using combined information from structures, energy predictions, and the known experimental information (binding and catalysis) for RASopathies to improve the distinction between RASopathies and cancer.

-quantitative rather than qualitative network differences determine the phenotypic outcome of RASopathy compared to cancer mutations, However, to have a true diagnostic power probably detailed analysis at each protein and individual position should be carried out.

Specific Points:

Comment 1: 1) The authors claim that RAS/MAPK proteins are involved in more pathways than others, and are widely expressed. Although this might well be true, the first conclusion suffers from selection bias: obviously, as an extremely well-studied signaling module, it is, all else being equal, likely to be reported to be involved in more pathways than others. Also, the latter is virtually tautological: components of this pathway belong to the same module, so as long as one is expressed, it is likely they all will be.

Reply 1: We see the reviewer's points that frequently-studied proteins will tend to have more interaction partners. The main message we wanted to transmit was in support of the hour-glass model where a small number of pathways are connected to a large number of receptors. We are confident this is the case for the RAS-MAPK-ERK module since we used information from the manually curated NetPath signaling database.

We have now added a sentence to the manuscript discussion as a note of caution that this bias could exist and also to make clear our message. (Inserted on page 15: "This is in support of the 'hour-glass model' of signaling, where a small number of pathway components are connected to a large number of receptors (Citri & Yarden, 2006); however we cannot exclude that some bias may exist as RAS/MAPK proteins are well studied and thus may be reported more frequently in pathway databases.) This potential bias, however, does not apply to the expression pattern since this comes from unbiased experimental work. Our statement that these proteins are widely expressed compared with other signalling proteins should hold. In this respect there is no tautology since we just wanted to point out that this pathway is present in almost all cells and embryological states and therefore mutations should be expected to have broad phenotypic characteristics.

Comment 2: 2) The authors conclude that RASopathy mutations have lower destabilizing energy values than do cancer mutations. Although this conclusion is likely to be true, the analysis selectively disregards some mutations, eventually using only a 'gold set' that had already been studied experimentally. Even then, whereas the destabilizing energy of cancer mutations as a class is clearly greater in Fig. 5, there is a considerable overlap between the two types of mutations. Hence, the utility of their conclusion is questionable: in other words, it is not clear that one could examine a new mutation by FoldX and classify it as likely cancer-causing or not. If the authors could, in fact do this (and demonstrate it convincingly), their findings would be of considerable interest for counseling and monitoring RASopathy patients about their cancer risk. However, without such ability, the claim that their approach has therapeutic or diagnostic implications does not seem warranted (and they certainly don't provide much discussion as to how/why it would be)

Reply 2: We appreciate the reviewer's concern and in the revised version of the manuscript we explain the two main aims of our study better. The first aim was to analyse what distinguishes RASopathy from cancer mutations on the molecular level. While they cannot be distinguished based on the localisation in different domains, it seems that the trend that was proposed earlier for some RASopathy based on experimental biochemical data is confirmed on a systems wide level for all RASopathy-related: In general the RASopathy mutations results in weaker de-regulation of the pathway, a finding that is confirmed by FoldX calculations. It is however true that the energy distributions of cancer and RASopathy mutation largely overlap, which probably precludes general diagnostic implications such as the prediction about whether RASopathic patients develop cancer.

The second aim was to analyse for a given protein all possible disease-causing mechanism (binding, stability, etc affected). We observe that even for the same gene, depending on the mutation, several disease-causing mechanisms exist. This is an important finding with potential therapeutic implications: several systems approaches nowadays aim to combine abundance with disease networks in a quantitative way, e.g. to understand why certain tissues or patients develop a disease and others not, despite the presence of the disease mutation in all tissues/ patients. Also drug therapy should be different if a mutation affects protein folding, or if it affects activation. In the first case as shown for transthyretin molecules binding to the folded state could rescue the phenotype, while in the second the active site could be the main target.

We explain now better the different aims of our study, in particular the therapeutic implication. (Inserted on page 18: “Our work shows that even for the same protein, depending on which disease mutation is affected, the effect on the network will be different, ranging from changes in abundance to changes in rate constants and affinities. This is a relevant finding for future system approaches that aim to combine tissue- or patient-specific protein abundances with disease networks in a quantitative way: in addition to protein abundances (node sizes in a network) and the presence and absence of interactions (qualitative edges or ‘edgetics’ in a network), quantitative effects on edges through changes in affinities and kinetic constants need to be considered (quantitative edges or ‘enedgetics’)”)

(Inserted/modified on page 17: “). It is, however, true that the energy distributions of cancer and RASopathy mutations largely overlap, which probably precludes general diagnostic implications such as the prediction about whether RASopathy patients develop cancer. This of course could be due to the fact that we are grouping mutations in different positions and proteins to have enough statistical power. This blurs specific position effects that should be considered to have diagnostic power. We have demonstrated for phenylketonuria and retinitis pigmentosa diseases that, when working with well characterized individual proteins and experimental biochemical data on binding or catalysis are available, FoldX-based quantitative stability prediction correlate well with the onset of the disease (Pey et al, 2007; Rakoczy et al, 2011). In fact we showed here that when focusing on a single protein (i.e. Ras) and using a toy network model in which experimental data for binding and catalysis were introduced (Gremer et al, 2011) effects, the distinction between RASopathy and cancer mutations can be improved. Thus to have true predictive power detailed analysis at each individual position should be performed and different factors considered. ”.)

Comment 3: 3) This model needs to be more clearly explained in the text. What are the starting conditions being modeled? The system starts with Ras-GDP, but what is the concentration of free GTP? After exchange followed by hydrolysis Ras returns to the GDP-bound state. Was the free GTP all hydrolyzed or is this the steady-state in the presence of GTP? The figure legend for panel C is exactly the same as B, although it appears to measure effector activation rather than Ras-GTP.

Reply 3: We explain the Ras activation model better now. Also, we provide an alternative model for Ras activation, where instead of activating Ras through a pulse (by recruiting the GEF to the membrane followed by GEF inactivation), we simulate steady state Ras activation under constant GEF and GAP activities, and effector binding conditions, which possibly better resembles the in vivo conditions of tumour formation.

Regarding the reviewers concern about free GTP and GDP: we do not model GTP or GDP binding to Ras explicitly. The reason is that Ras is unfolded in the absence of a bound nucleotide. Also the model assumes that the cellular level of GTP is much higher than the Ras concentration and therefore it is not explicitly introduced in the model since it will not be rate limiting (this is what is done in all RAS-MEK-EKR models; Kholodenko *et al*, 1999; Kiyatkin *et al*, 2006).

Regarding the figure legend: We have corrected the text for the figure legend.

(Inserted on page 24/25: Network simulations...)

Comment 4: Minor concern: "RAS/MAPK proteins are involved in more pathways than other proteins": While this is probably true, this trend must also reflect the fact that these 12 proteins are disproportionately well studied with more publications in the literature to implicate them in more

pathways.

Reply 4: This comment is answered in comment 1/reply 1.

Comment 5: 4) *The authors state: "We predicted for some Ras mutants that in addition to preventing Ras GTP hydrolysis, they also decrease binding to the Ras activating GEF protein. More interestingly, the compensatory energies are stronger for RASopathies compared to cancer mutations, which could explain the mild phenotype of RASopathy compared to cancer mutations." Yet by comparing the ratios for FoldX values for RasGAP versus RasGEF in cancer versus RASopathy, it is not clear to me that there is any difference. GEF interactions are more destabilized by RASopathy mutations than are cancer mutations, but so are GAP interactions. Has a statistical analysis of the GAP/GEF energy ratios been performed?*

Reply 5: We explain this better now. We meant with compensating that both GEF and GAP binding are SIMILARLY affected. However, as GEF is activating and GAP deactivating, the net result on Ras activation will be compensatory. On request by reviewer 3, mentioning that compensatory mutations are not so novel (Gremer et al), we moved the main figure 6 into the supplement and only mention that we also see compensatory effects. As we have also have removed duplicated amino acid positions (because of H, H, NRas isoforms) and Q61 mutations (as they are probably not reliable because of larger structural changes), the number of cases is too low for performing a statistical analysis. We now integrate the effect of the compensatory mutations in to our toy model to show their important contribution to the final level of activated ERK in RASopathy and cancer mutations.

(Changed on page 12/13)

Comment 6: 5) *The model in Fig. 7 should be more clearly explained in the text. For example, what are the starting conditions? The system starts with RAS-GDP, but what is the concentration of free GTP? After exchange, followed by hydrolysis, RAS returns to the GDP-bound state. Was the free GTP all hydrolyzed or is this the steady-state in the presence of GTP? Also, the legend for panel C is exactly the same as B, although it appears to measure effector activation rather than RAS-GTP.*

Reply 6: On request by all three reviewers we explain this section much better now.

Regarding the figure legend: We have corrected the text for the figure legend (however, in fact the figure has changed). Regarding free GTP, please see reply 3.

(inserted on page 24/25)

Comment 7: 6) *It is unclear why the authors include RASA1 as a RASopathy gene; most would not. Also unclear is why they fail to consider RIT1, which was reported to be a major remaining gene for NS last year by Aoki et al.*

Reply 7: We appreciate the suggestion to carefully analyse which RASopathy genes to include in our analysis. We now base this decision on the latest review on RASopathies "the RASopathies, by K. A. Rauen in Annu. Rev. Genom. Human. Genet, 2013). This review includes RASA1, and additionally CBL and SHOC2. The two genes have been integrated into our analysis in the revised version of the manuscript. We have also now added RIT1; we apologize we missed it, but the publication came out last summer, essentially when we had already finished the structural analysis and energy calculations.

Comment 8: 7) *Why don't the authors include colon cancer as a disease in which KRAS mutations are common (~50% of cases)?*

Reply 8: Sorry, that was a mistake. We have now added this connection in Figure 1B.

Comment 9: 8) *Similarly, it is not clear why the authors state that MEK mutations have not been observed in cancer. In fact, N-terminal MEK mutations have been reported in lung cancer and as resistance mutations in melanoma.*

Reply 9: We thank the reviewer for these comments. We have now included MEK mutations and performed the energy calculations.

Comment 10: 9) *Many of the references are incorrect and/or inappropriate. Some times primary papers are cited that do not have the data stated, other times, reviews are cited that are not always up to date. Overall, the referencing is quite selective.*

Reply 10: We thank the reviewer for pointing this out, and apologize for the incomplete and partly incorrect references used. We have performed a better literature search, and included new publications and reviews.

Reviewer #2:

Summary

In this manuscript the authors aim to explain the differences between the mutations in the Ras/MAPK genes that lead to the development of cancer or RASopathies. Interestingly, different mutations in the same gene, and in some cases even in the same residue, result in different outcomes. Thus, the authors try to shed light on an intriguing and relevant question that is difficult to address by using only experimental approaches. This manuscript by Kiel & Serrano analyses 508 different missense mutations in the 12 genes related to RASopathies and cancer using a computational approach. By combining protein network data with structure-energy based predictions the authors conclude that quantitative rather than qualitative differences in Ras dependent signalling pathway determine the phenotype. The authors also develop a simple mathematical model to simulate the effects of some of the different mutations in the Ras/MAPK network to further support their conclusions.

The authors conclude that the systemic approach used in the current study that integrates structure-energies predictions and predictive mathematical modelling is a valid method to study the different effects on a signalling network cause by different mutation in the same proteins

General remarks.

The work described by Kiel and Serrano is in general well supported by the data presented. The systematic computational analysis described in this study can be applied to other signalling networks. It also demonstrates the importance of considering protein structure in order to get a better understanding of the molecular mechanisms how signalling networks work.

Importantly, the data presented here helps to understand the different role of Ras/MAPK mutants in cancer and RASopathies. Thus, this manuscript extends our knowledge on disease-causing mechanisms of RASopathies. The authors present a thorough analysis of the mutations in the 12 genes responsible using a combination of known data curated from the literature and several databases with new computational approaches. They are able to explain different phenotypes observed for different mutations through network analysis and protein structure. Their conclusions from these studies are supported by good correlation with experimental data from the literature (Table S6) and predictions of the mathematical model generated. It must be noted that this computational approach allows us to get insight into the biological processes without the need to perform additional experiments. It is, however, dependent on the availability of protein structures (in this case 11 out of 12, but some relevant regions are missing).

This work can be used to estimate effects of new, uncharacterised mutations. The manuscript is interesting for a wide audience including structural and computational biologist but also researches

interested in signal transduction. Overall this manuscript is relevant for the audience of this journal.

Specific comments and suggestions through manuscript.

Major points.

Comment 11: *Although the manuscript fulfils the criteria for its publication on MolSysBiol this reviewer thinks that some editing is necessary before its publication. There are numerous semantic and grammatical mistakes in the text (see below), and the text will benefit from rewriting in a more concise and careful way. The introduction is rather long and some of the points made are not necessary. Specifically, the description of some of the results in this part can be shortened since they are explained in the results and discussion. Also, I would suggest the author to consider editing the first paragraph of the classification of disease mutations and structural coverage. It reads as introduction within the results section and may be better to be included in the actual introduction.*

Reply 11: We agree with the Reviewer's suggestion and have shortened the introduction. We have deleted the first result chapter on the disease. However, we disagree with the reviewer regarding the classification of disease mutations. We prefer to keep this in the results sections as this is crucial to our structure-based classification approach and needs to go together with figure 3.

The language has been corrected by a native speaker.

Comment 12: *I also would suggest moving Figures S5 and S6 into the main text. These figures are describing the domain analysis of missense mutations in RASopathies and cancer, which would deserve to be shown in the main text rather than as supporting material.*

Reply 12: We thank the reviewer for this suggestion. We include Figures S5 and S6 in the main text now.

Comment 13: *The oncogenic RasG12V mutation is known to lock Ras in an active conformation caused by the loss of GTPase activity. However, in the mathematical model predictions RasV12T is shown to come back to an inactive state. How do the authors explain this? Activation of wt Ras proteins has been shown to be important for oncogenic Ras transformation (Jeng HH et al. 2012, Nature Communications; Young A, 2013, Cancer Discovery; Matallanas et al. Molecular Cell, 2010). Does the model consider total Ras activation including the other Ras isoforms and/or the wt allele of the mutated isoform? Is this is what is shown in this prediction?*

Reply 13: The Ras G12V is not locked in the active conformation because of a 'loss of GTPase activity', but because of a loss in GAP-stimulated GTPase activity (Ahmadian *et al.*, 1997). The intrinsic (slow) GTPase activity is not or only minor changed. Therefore RasT levels go slowly down when the initial GEF stimulus is stopped. We explain these differences in intrinsic and GAP-stimulated GTPase activity now better in the methods section.

We explain the model better now (see also comment 3). In fact, inspired by reviewer 3 we generated a different model for Ras activation, where instead of activating Ras through a pulse (by recruiting the GEF to the membrane followed by GEF inactivation), we simulate steady state Ras activation and effector binding under constant GEF and GAP activities, and effector binding conditions, which possibly better resembles the in vivo conditions of tumour formation.

We are not considering a wt allele.

(inserted on page 24/25)

Comment 14: *The Ras Q61 mutant is mentioned the first time in the discussion section, why is it not*

used in the analysis or as prediction?

Reply 14: There are larger structural (backbone) changes for this mutation in Ras (seen in the X-ray structure of the Q61L mutation, Buhrman *et al*, 2007). The FoldX design algorithm fails if a mutation introduced larger structural backbone changes, and therefore was thus excluded from the analysis (but included in Table S1). We mention the limitation of FoldX with respect to modeling backbone changes in the revised version of the manuscript.

(inserted on page 22: “The FoldX algorithm allows predictions of mutational affect for any of the 20 natural amino acids, but it does not allow any backbone changes. Thus, for example, predictions for RasQ61 mutations may not be reliable, as the structure of the Ras Q61L mutation has been solved and it shows larger structural changes (Buhrman et al, 2011).”)

Minor points

Comment 15: *Page 4: SPRED1 prevents RAF1 activation by Ras, but is NOT an inhibitor of the Ras-RAF1 interaction.*

Reply 15: We have corrected this.

(Inserted on page 4: “SPRED1 prevents RAF1 activation by Ras”)

Comment 16: *Page 8: Reference Goh et al. requires re-formatting*

Reply 16: We have reformatted all references according to MSB style.

Comment 17: *Table S1: Formatting in the download is very poor. Please make sure that this corrected in the final version.*

Reply 17: This table should be a xls-file in the final version; maybe the reviewer received a pdf file for reviewing the manuscript.

Comment 18: *Page 9: Why did the authors choose this database over others? Wrong reference: Kandasamy, K. and Mohan, SS. et al. (2010) NetPath: A public resource of curated signal transduction pathways. Genome Biology. 11:R3 Reference in figures S3 and S4 are missing (Geiger et al 2013, Su et al 2004) and this reviewer failed to find them with the information provided in PubMed.*

Reply 18: We used Netpath as it is a manual curated database of signaling pathways.

We have corrected the reference for NetPath. We have added the references to figures S3 and S4.

Comment 19: *Page 11: SPRED1 is used here as an example but the authors say earlier (fig 1B) that it has no cancer-related mutations. Please clarify.*

Reply 19: We apologize for this confusion. In fact Figure 1B and 1C contradict each other as figure 1B has been done with the main cancer based on Goh et al as the state-of-the art (and there SPRED1 was not associated to cancer), and figure 1C using all cancer mutations in the COSMIC database

(where cancer mutations were identified). We have changed figure 1B in the revised version of the manuscript so that it reflects our mutant data in table S1.

Comment 20: Page 11: "An exception is SPRED1, but here the total number of mutations is low and probably not statistically significant." What does "probably not statistically significant" mean? Statistical significance can be tested and a result can be significant or not.

Reply 20: We have updated the number of cancer mutations for SPRED1 and the other proteins in the revised version. We have deleted the sentence that for SPRED1 the total number of mutations are low.

Comment 21: Page 11: "For MAP2K1 and MAK2K2 no cancer mutations are identified." This is incorrect. See *Cancer Res.* 2008 Jul 15;68(14):5524-8. doi: 10.1158/0008-5472.CAN-08-0099. Novel MEK1 mutation identified by mutational analysis of epidermal growth factor receptor signaling pathway genes in lung adenocarcinoma.; Marks JL, Gong Y, Chitale D, Golas B, McLellan MD, Kasai Y, Ding L, Mardis ER, Wilson RK, Solit D, Levine R, Michel K, Thomas RK, Rusch VW, Ladanyi M, Pao W. *Cancer Res.* 2011 Aug 15;71(16):5535-45. doi: 10.1158/0008-5472.CAN-10-4351. Epub 2011 Jun 24. Identification of the MEK1(F129L) activating mutation as a potential mechanism of acquired resistance to MEK inhibition in human cancers carrying the B-RafV600E mutation. Wang H, Daouti S, Li WH, Wen Y, Rizzo C, Higgins B, Packman K, Rosen N, Boylan JF, Heimbroad D, Niu H.

Reply 21: We thank the reviewer for pointing this out, and apologize for having overlooked this literature. We now included the MEK cancer mutations and performed the FoldX energy calculations.

We have also included three new proteins to the analysis after careful literature search.

Comment 22: This reviewer would suggest to re-format figure 3 by splitting it into parts A and B, as the second part explains the different classes and the first part the procedure (and is not referred to in text). The structures in the second part are referred to as pdb-id in the figure while their names are used in corresponding text. Please relate them to each other.

Reply 22: We appreciate the suggestion and have splitted figure 3 in panels A and B. We also relate pdb entries to protein names in the figure in the revised version of figure 3.

Comment 23: Page 16: Figure 6D+E are merged in text and legend

Reply 23: We have fixed this in the revised manuscript text.

Comment 24: Page 17: Legend for figure 7C is wrong (identical to 7B)

Reply 24: We have fixed this in the revised manuscript text.

(inserted on page 34/35)

Comment 25: Please correct the numerous grammatical errors and wrong use of words, e.g.:

Page 4: "RAS1 encodes for the p120..."

Page 4: "Binding of 14-3-3 proteins have been shown..."

Legend to Figure 2A "...the colour and size of the nodes represents the number of signalling pathway

they predicate in (see legend)." - this sentence contains two errors, i.e. represents instead of represent, and predicate instead of participate.

Page 16: "...is only minor increased..."

Reply 25: We have fixed all of these instances in the revised manuscript text.

Comment 26: *Page 19: "We show that the majority of the mutations involved in the twelve RASopathy proteins that we could model and resulted in significant energy changes (73%)."*

Reply 26: We have corrected this sentence in the revised manuscript text.

(corrected on page 16: "We show that the majority of the mutations involved in the 15 RASopathy proteins we could model resulted in significant energy changes when modelled on the available crystal structures (>0.8 kcal/mol; 68%)")

Comment 27: *Page 20/21: BRAF 600E is not kinase impaired. Beside, mutations in BRAF and KRAS are usually mutually exclusive. Please correct.*

Reply 27: We have corrected this in the revised manuscript text.

(corrected on page 16: "For example, BRAF 600E mutations treated with a kinase inhibitor can still activate the MEK-ERK pathway through heterodimerization with RAF1 (Heidorn et al, 2010).")

Comment 28: *The authors state that "with the exception of H-Ras and MAP2K2, all RASopathy genes are embryonic lethal when knocked out [in mice]". This is not true. N-Ras and K-Ras4A mice are viable and only K-Ras4B is embryonic lethal (Plowman et al 2003, and Umanoff 1995). Please correct.*

Reply 28: We thank the reviewer for pointing this out, and apologize for this mistake. It is fixed in the revised manuscript.

(corrected on page 6: " In fact, many of the RASopathy genes are embryonic lethal when knocked out, and thus are essential disease genes (Dickerson et al, 2011). Exceptions are HRAS, NRAS, MAP2K2; KRas4 mice are viable and only KRas4B is embryonic lethal (Umanoff et al, 1995; Plowman et al, 2003). In case of CBL, only deleting both, CBL and CBLB results in embryonic lethality (Naramura et al, 2002). No information on embryonic lethality was found for Rit1.")

Comment 29: *The authors state that " RAF1 while producing the same RASopathies, it is not involved in tumour formation". This is not true. Although rare, mutations on this gene have been described in the literature (Zebisch et al 2006). Furthermore, the involvement of this protein in cancer is also suggested by the re-activation of ERK in patients with mutant B-Raf treated with vemurafenib which seems depends on Raf-1. Please correct and clarify what you mean.*

Reply 29: We thank the reviewer for pointing this out, and apologize for this mistake. We include FoldX prediction for Raf-1 cancer mutations in the revised version, and the sentence has been deleted.

Comment 30: *There is inconsistency in the nomenclature of the proteins i.e. both Raf1 and CRaf are used.*

Reply 30: We name it RAF1 throughout the revised manuscript.

Reviewer #3:

The paper by Kiel and Serrano is a systems biology approach in order to better understand RASopathy mutations and their relation to cancer mutation. RASopathies like Noonan, Leopard or CFC syndrome are developmental diseases mutated in genes that can be aligned in a single pathway that goes from activated RTKs downstream to activation of Erk and involves mainly Ras and its regulators and effectors. What makes such a study particularly interesting is that the same genes are also mutated in cancer which begs the question why these mutations don't lead to cancer (which some do, see below).

The authors first collect and order the large number of patient mutations and make a wonderful representation and categorization of the diseases. They show that these proteins participate in early to late developmental pathways which is why such patients live. They observe that RASopathy and cancer mutations cannot be distinguished on the bases of the type of domains which are mutated. They then use the program FoldX to calculate the change in energy for those proteins and/or domains for which a three-dimensional structure is available, the others are analyzed according to the degree of conservation and the energy change expected for such a mutation (be it on the surface). By FoldX and structural considerations (where structures are available) the mutations are then categorized by the type of effect they might have on activity, interaction with regulators and effectors and on folding. Of the 339 mutants, 82% have significant energy changes higher than 0,8 kcal. 93 mutants do not show any significant energy changes and could represent proteins that have a defect in interaction with partner proteins. The authors claim that cancer mutations show larger energy changes than RASopathies.

In a more detailed analysis the authors model the effect of the individual Ras mutations on the Ras-MApK pathway, Relying on a set of biophysical measurements by Gremer et al, they come to the conclusion that Ras activation-inactivation is between the wt and oncogenic G12V Ras, supporting the notion that Ras activation in the diseases is less severe and sustained than in RASopathies.

Overall this is a nice summary and investigation that broadens our overall understanding of the diseases which needs however some explanations, clarifications and corrections.

Comment 31: *The separation of mutation into the two categories is not so straightforward, as kids with RASopathies do sometimes develop cancer (i.e JMML) The data maybe difficult to come by, but should at least be discussed, possibly in terms of severity of the disease.*

Reply 31: The reviewer makes an important point that we have not addressed well enough in the manuscript (and that is also related to comment 32). We have included a discussion on this topic in the revised version of the manuscript. In fact this is also related to the energy distribution of RASopathy and cancer mutations in figure 5, which despite a statistical significant trend, show a large overlap. We discuss that indeed we could not establish an energy threshold to discriminate mutation effects when looking globally at all positions and proteins, but could work when looking at individual positions (see Kiel et al., JMB Rhodopsin) and enough data is available (see also Fig 6, the output of the toy model for different Ras mutants).

(inserted on page 17/18: "It is, however, true that the energy distributions of cancer and RASopathy mutations largely overlap, which probably precludes general diagnostic implications such as the prediction about whether RASopathy patients develop cancer. This of course could be due to the fact that we are grouping mutations in different positions and proteins to have enough statistical power. This blurs specific position effects that should be considered to have diagnostic power. We have demonstrated for phenylketonuria and retinitis pigmentosa diseases that, when working with well characterized individual proteins and experimental biochemical data on binding or catalysis are available, FoldX-based quantitative stability prediction correlate well with the onset of the disease (Pey et al, 2007; Rakoczy et al, 2011). In fact we showed here that when focusing on a single protein (i.e. Ras) and using a toy network model in which experimental data for binding and catalysis were introduced (Gremer et al, 2011) effects, the distinction between RASopathy and cancer mutations can be improved. Thus to have true predictive power detailed analysis at each individual position should be performed and different factors considered. For example it has also been proposed that Raf-1 could increase the intrinsic GTP hydrolysis on Ras, which has been associated to different

transforming activities of mutations at position Q61 of Ras (Buhrman et al, 2007). Other effects, such as the preferential expression of isoforms during development or even different localization dynamics (Chandra et al, 2012) should be taken into account to explain why Costello syndrome mutations that harbour a G12X mutation in HRAS do not show up frequently in cancer (although Costello syndrome patients do develop tumors more frequently; Gripp & Lin, 2012), while KRAS G12X mutations are frequently involved in cancer.. “)

Comment 32: It would be worth mentioning/discussing Leopard mutations that harbor a G12X mutation in H-Ras, which do in principle not show as cancer, why?

Reply 32: We assume that the reviewer means Costello syndrome mutants in H-Ras? Indeed that is something that we cannot explain through our FoldX analysis, e.g. why KRAS G12X mutations lead to cancer and HRAS G12X mutations lead to syndromes (and to a smaller extent to cancer). We discuss this now (which is related to comment 31), and it suggests that KRAS G12X is more important during development and that's why cancer mutations are not tolerated, while for HRAS G12X mutations are tolerated, possibly because HRAS is less expressed in the developing embryo.

(inserted on page 17/18: “see comment 31”)

Comment 33: I also have a slight problem with extrapolating energy changes to changes in activity/affinity ect, which I am certain the authors are aware of, but don't discuss in due form. A good example is G12V (Table S6). They show a severe destabilization by FoldX, where in fact this mutants is quite normal in terms of its biochemical properties, including binding to GAP, except that it cannot be activated because arginine cannot swing into the active site.

Reply 33: We see the Reviewer's points here and have endeavoured to clarify each of those concepts better in the revised manuscript method section. Indeed RasG12V mutations when modelled on the Ras structure alone are not destabilizing and folding is not expected to be affected (we highlight this better in the method section and table S1). However, when we model RasG12V mutations in the Ras-GAP X-ray structure, it essentially results in van der Waals clashes with the Arg from the GAP that does the catalytic job. The reason is that FoldX does not allow backbone or loop changes and therefore in loops one could get large energy changes that will not unfold the protein, but will change loop conformation destroying activity.

We explain these concepts better in the revised version of the manuscript.

(inserted on page 22:” The FoldX algorithm allows predictions of mutational affect for any of the 20 natural amino acids, but it does not allow any backbone changes. Thus, for example, predictions for RasQ61 mutations may not be reliable, as the structure of the Ras Q61L mutation has been solved and it shows larger structural changes (Buhrman et al, 2011). This is especially important for mutations in loops (for example when modelling Ras G12V in complex with GAP); in this case introducing mutations in the crystal structure results in van der Waals clashes and therefore large energy changes, which, however, will not unfold the protein, but will change loop conformation preventing GTP hydrolysis through GAP protein.”

Comment 34: My greatest problem is with the modelling of Ras activation, needs more explanation: What happens at time point zero, and what is the exchange factor reaction that is modeled here, what are the rates being used. The rates experimentally determined were just k-obs, for a certain concentration of Ras and regulators and effectors. In the real world, the GEF would be SOS, and SOS is a very weak enzyme that in fact needs to be activated by a feedback (feed forward) mechanism. For the second order reactions one would need concentrations which are given but not explained where they come from (Suppl. Table 3 and 4). They are unlikely to come from expression levels given in the Tables, or do I miss something here.

Reply 34: We understand the reviewer's point and have considerably improved the description of the model in the method section and we corrected some errors. While the rates for intrinsic GTP-hydrolysis and intrinsic GDP dissociation are concentration independent, the reviewer is right that k_{obs} values depend on a certain amount of Ras and regulators and cannot be used directly for rates in reaction 5 and 7 of the model. We have changed the model by integrating the change in k_{obs} compared to WT to the rate for these reactions used in previously published mathematical models of Ras activation and deactivation.

We have further clarified where the abundances in the model come from (average from previously published models and our own in solution abundances measured in different cell types; Kiel *et al* JPR, 2014)

Regarding the comment on the "rate constants in the real world": we appreciate the reviewers concern; however we argue that every model needs simplifications. We decided to use in-solution rate constants as published by Gremer *et al*. However, the reviewer is right that in the 'real world' proteins are localized, often through the use of multiple domains, which will increase the effective concentration and/or affinities between species. We could have included the positive feedback on Sos in our model, but we decided not to do so in order to directly integrate and compare 5 experimental in solution rate constants for different cancer and RASopathy mutations. In addition, none of the Ras mutations are localized in the distal Ras binding site (the region that binds to SOS1 in the positive feedback). It is very likely that other proteins (effector/ GAPs) are also localized with an effect on the effective concentration. As modelling reality is difficult, we wanted to keep this model simple and consider in solution rate constants only.

We explain the assumptions and simplifications of our model in the revised version of the manuscript.

The initial concentrations of species in the model (i.e. total Ras [=sum of H, K, and N-Ras], GEF [=Sos1], GAP [=RASA1], and EFF [=sum of RAF1 and BRAF]) were averages based on experimentally determined protein abundances in three mammalian cell lines (Kiel *et al*, 2014) (see Supplementary table S6)

Comment 35: *And why is Ras activation eventually going down to zero? In reality one would expect, at least for the mutants, them to be at a certain level above zero, that's why they cause disease. Fig7C shows indeed that the activation of effectors is more difficult to model explaining that the model needs some modification. I partially agree with the authors that this might be due to the inability to model Raf activation. This should at least be discussed.*

Reply 35: We see the Reviewer's point and have done the following changes: In order to avoid entering into the complicated mechanism of effector activation, we only show the RasT-effector complex formation, without having an effector activation step in the model.

Ras activation eventually goes down to zero as we stop the GEF stimulus. Then, Ras activation state will mostly depend on the intrinsic GTP hydrolysis and intrinsic GDP exchange. As the intrinsic GDP exchange rate is two orders of magnitude smaller compared to the GTP hydrolysis, the active Ras will decrease close to zero.

However, as this stimulus-dependent activation in the model is probably artificial (although in many experimental studies done that analyse the effect of Ras/Raf cancer mutations on ERK activation), we have now modified the model to a steady-state model, where instead of activating Ras through a pulse (by recruiting the GEF to the membrane followed by GEF inactivation), we simulate steady state Ras activation under constant GEF and GAP activities, and effector binding conditions, which possibly better resembles the in vivo conditions of tumour formation.

Comment 36: *The authors mention mutants G60R which I cant see in Fig. 7b*

Reply 36: We apologize that the legend was a little bit small (however G60R was there). We have replaced the figure by new simulations and the labels are bigger now.

Comment 37: On p.16, they talk about compensatory mutations, where GTP hydrolysis defect would be compensated by a decrease in SOS binding, which would not be compensatory, and this is in fact mentioned/shown in Fig. 6

The real compensatory mutations have been discovered by Gremer et al., and that should be mentioned on p16, already, in that same chapter.

Reply 37: Although it was mentioned in the chapter, we mention this reference now in the beginning “In agreement with previous experimental observations (Gremer et al., 2011), we also identified compensatory effects of Ras mutants with respect to binding to Sos1.”

Comment 38: P19, it is mentioned that Rasopathies arise from mutations in more than one gene, which is misleading, should be rephrased, there is one gene-one disorder in each patient.

Reply 38: We have changed the sentence.

(inserted on page 15: “RASopathies are a class of disorders with overlapping disease symptoms that arise from mutations in different genes (locus heterogeneity)”

Comment 39: P19, we found that most of the..., did the authors do the expression tests???

Reply 39: We mention that this claim is based on published transcript and protein expression levels (figure S3,S4). We mention the references to those papers at this point in the revised version of the manuscript.

(Inserted on page 15).

Comment 40: P21, why should BRAFV600E not activate MEK-ERK?

Reply 40: We apologize. “kinase-impaired V600E mutations” was worded totally wrong, it should say: “BRAF V600E mutations treated with a kinase-inhibitor”

(Corrected on page 16).

References

Ahmadian MR, Stege P, Scheffzek K, Wittinghofer A (1997) Confirmation of the arginine-finger hypothesis for the GAP-stimulated GTP-hydrolysis reaction of Ras. *Nat Struct Biol* **4**: 686-689

Buhrman G, Wink G, Mattos C (2007) Transformation efficiency of RasQ61 mutants linked to structural features of the switch regions in the presence of Raf. *Structure* **15**: 1618-1629

Dickerson JE, Zhu A, Robertson DL, Hentges KE (2011) Defining the role of essential genes in human disease. *PLoS One* **6**: e27368

Kholodenko BN, Demin OV, Moehren G, Hoek JB (1999) Quantification of short term signaling by the epidermal growth factor receptor. *J Biol Chem* **274**: 30169-30181

Kiyatkin A, Aksamitiene E, Markevich NI, Borisov NM, Hoek JB, Kholodenko BN (2006) Scaffolding protein Grb2-associated binder 1 sustains epidermal growth factor-induced mitogenic and survival signaling by multiple positive feedback loops. *J Biol Chem* **281**: 19925-19938

Naramura M, Jang IK, Kole H, Huang F, Haines D, Gu H (2002) c-Cbl and Cbl-b regulate T cell responsiveness by promoting ligand-induced TCR down-modulation. *Nat Immunol* **3**: 1192-1199

Plowman SJ, Williamson DJ, O'Sullivan MJ, Doig J, Ritchie AM, Harrison DJ, Melton DW, Arends MJ, Hooper ML, Patek CE (2003) While K-ras is essential for mouse development, expression of the K-ras 4A splice variant is dispensable. *Mol Cell Biol* **23**: 9245-9250

Umanoff H, Edelman W, Pellicer A, Kucherlapati R (1995) The murine N-ras gene is not essential for growth and development. *Proc Natl Acad Sci U S A* **92**: 1709-1713

Acceptance letter

25 March 2014

Thank you again for sending us your revised manuscript. We are now satisfied with the modifications made and I am pleased to inform you that your paper has been accepted for publication.

Thank you very much for submitting your work to Molecular Systems Biology.

Reviewer #3:

I am happy with the changes that have been made